# CYRI-B-mediated macropinocytosis drives metastasis via lysophosphatidic acid receptor uptake

Savvas Nikolaou[1], Amelie Juin[1], Jamie A Whitelaw[1], Nikki R Paul[1], Loic Fort[1†], Colin Nixon[1], Heather J Spence[1], Sheila Bryson[1], Laura M Machesky[1,2*‡]

[1]CRUK Scotland Institute, Switchback Road, Bearsden, Glasgow, United Kingdom; [2]Institute of Cancer Sciences, University of Glasgow, Glasgow, United Kingdom

**\*For correspondence:**
lmm202@cam.ac.uk

**Present address:** †Department of Cell and Developmental Biology, Vanderbilt University, Nashville, United States; ‡Department of Biochemistry, University of Cambridge, Cambridge, United Kingdom

**Competing interest:** The authors declare that no competing interests exist.

## Abstract

Pancreatic ductal adenocarcinoma carries a dismal prognosis, with high rates of metastasis and few treatment options. Hyperactivation of KRAS in almost all tumours drives RAC1 activation, conferring enhanced migratory and proliferative capacity as well as macropinocytosis. Macropinocytosis is well understood as a nutrient scavenging mechanism, but little is known about its functions in trafficking of signalling receptors. We find that CYRI-B is highly expressed in pancreatic tumours in a mouse model of KRAS and p53-driven pancreatic cancer. Deletion of *Cyrib* (the gene encoding CYRI-B protein) accelerates tumourigenesis, leading to enhanced ERK and JNK-induced proliferation in precancerous lesions, indicating a potential role as a buffer of RAC1 hyperactivation in early stages. However, as disease progresses, loss of CYRI-B inhibits metastasis. CYRI-B depleted tumour cells show reduced chemotactic responses to lysophosphatidic acid, a major driver of tumour spread, due to impaired macropinocytic uptake of the lysophosphatidic acid receptor 1. Overall, we implicate CYRI-B as a mediator of growth and signalling in pancreatic cancer, providing new insights into pathways controlling metastasis.

## Editor's evaluation

This important study combines in vivo and in vitro models to characterise the complex role of CYRI-B, an interactor of the small GTPase Rac1, in controlling pancreatic cancer progression towards a higher proliferative and metastatic stage. The authors demonstrate that CYRI-B reduces the typical hyperactivation of Rac1 in early stages of tumor progression; subsequently, CYRI-B mediates internalization of lysophosphatidic acid receptor 1 (LPAR1) uptake through macropinocytosis, thus regulating chemotactic migration of cancer cells towards lysophosphatidic acid (LPA). This work, based on convincing evidence, will be of broad interest to cell biologists and the signalling research communities.

## Introduction

Pancreatic ductal adenocarcinoma (PDAC) is highly metastatic with low survival rates and few treatment options. PDAC is thought to arise from precancerous non-invasive pancreatic intraepithelial neoplasms (PanINs) classified as PanIN1–3 depending on the molecular and histological characteristics (*Hruban et al., 2001*; *Hruban et al., 2007*). PanINs arise from acinar cells that undergo acinar to ductal metaplasia changes (*Wang et al., 2010*) and as mutations accrue, PanINs progress to full PDAC in which angiogenesis, infiltration of stromal cells, and invasion of the basement membrane occur as tumours progress. Metastasis is a complex process and the current gold standard mouse model of metastatic PDAC - the KPC (KRAS$^{G12D}$, p53$^{R172H}$, Pdx-1-Cre) (*Hingorani et al., 2005*) recapitulates

**eLife digest** Pancreatic cancer is an aggressive disease with limited treatment options. It is also associated with high rates of metastasis – meaning it spreads to other areas of the body. Environmental pressures, such as a lack of the nutrients metastatic cancer cells need to grow and divide, can change how the cells behave. Understanding the changes that allow cancer cells to respond to these pressures could reveal new treatment options for pancreatic cancer.

When nutrients are scarce, metastatic cancer cells can gather molecules and nutrients by capturing large amounts of the fluid that surrounds them using a mechanism called macropinocytosis. They can also migrate to areas of the body with higher nutrient levels, through a process called chemotaxis. This involves cells moving towards areas with higher levels of certain molecules. For example, cancer cells migrate towards high levels of a lipid called lysophosphatidic acid, which promotes their growth and survival.

A newly discovered protein known as CYRI-B has recently been shown to regulate how cells migrate and take up nutrients. It also interacts with proteins known to be involved in pancreatic cancer progression. Therefore, Nikolaou et al. set out to investigate whether CYRI-B also plays a role in metastatic pancreatic cancer.

Experiments in a mouse model of pancreatic cancer showed that CYRI-B levels were high in pancreatic tumour cells. And when the gene for CYRI-B was removed from the tumour cells, they did not metastasise. Further analysis revealed that CYRI-B controls uptake and processing of nutrients and other signalling molecules through macropinocytosis. In particular, it ensures uptake of the receptor for lysophosphatidic acid, allowing the metastatic cancer cells to migrate.

The findings of Nikolaou et al. reveal that CYRI-B is involved in metastasis of cancer cells in a mouse model of pancreatic cancer. This new insight into how metastasis is controlled could help to identify future targets for treatments that aim to prevent pancreatic cancer cells spreading to distant sites.

multiple features of the human disease (*Hwang et al., 2016*). Cytoskeletal and migration-associated proteins have been associated with aggression and metastasis in PDAC both in human patient transcriptomes (*Bailey et al., 2016*) and in the KPC mouse model (*Juin et al., 2019*; *Li et al., 2014*), suggesting avenues to pursue against metastatic spread.

Downstream of active KRAS, hyperactivation of the small GTPase RAC1 drives proliferation and cytoskeletal remodelling in PDAC and other cancers. Deletion of RAC1 in a KRAS-driven mouse model of PDAC delayed tumour onset, reduced PanIN lesions, and improved survival (*Heid et al., 2011*; *Wu et al., 2014*). This led to the conclusion that dysregulation of RAC1 control of epithelial polarity by active KRAS drives acinar to ductal metaplasia and accelerates tumourigenesis (*Heid et al., 2011*). RAC1 regulates polarity and migration via Scar/WAVE-Arp2/3 control of actin dynamics at cell-cell contacts and at the cell leading edge. Additionally, coordinated RAC1 activation and deactivation are important in macropinocytosis, an actin-driven process whereby cells engulf extracellular substances via large cup-shaped protrusions of the plasma membrane (*Egami et al., 2014*). Extracellular stimulation of cell surface receptors, such as tyrosine kinase or G-protein-coupled receptors, can trigger macropinocytosis via RAC1 and the Scar/WAVE complex (*Buckley and King, 2017*). Tumours are frequently starved for amino acids and other nutrients and macropinocytosis is a major way for PDAC tumours to take up proteins, lipids, and cell debris from their environment (*Commisso et al., 2013*; *Hobbs and Der, 2022*; *Kamphorst et al., 2015*; *Puccini et al., 2022*; *Yao et al., 2019*). Macropinocytosis also provides cells with a mechanism for internalisation of signalling receptors (*Clayton and Cousin, 2009*; *Le et al., 2021*; *Stow et al., 2020*), but whether this has consequences for tumour progression is unknown.

Metastasis is a complex process, involving cells breaching through tissue barriers, migrating and settling in distant sites in the body such as the liver, lungs, and peritoneal cavity (*Nikolaou and Machesky, 2020*). Chemotaxis is thought to be a key driver of metastasis and pancreatic cancer cells migrate towards lysophosphatidic acid (LPA) both in vitro and in vivo, contributing to metastasis (*Juin et al., 2019*; *Papalazarou et al., 2020*). LPA is a serum-derived chemotactic factor and was previously found to be consumed by melanoma and PDAC cells creating self-generated gradients contributing to metastasis (*Juin et al., 2019*; *Muinonen-Martin et al., 2010*).

Recently, the CYRI-B protein (**Cy**fip-related **R**AC1-**i**nteracting protein B, formerly known as Fam49-B) was discovered to interact with RAC1 and enhance leading edge actin dynamics by negatively regulating activation of the Scar/WAVE complex (*Fort et al., 2018*). Scar/WAVE is a pentameric complex that interacts with both RAC1 and Arp2/3 complex and triggers actin assembly in lamellipodia (*Insall and Machesky, 2009*). Depletion of CYRI-B in cultured cells enhanced lamellipodia stability, but did not impair migration speed (*Fort et al., 2018*). CYRI proteins also play an important role in macropinocytosis, via the RAC1-Scar/WAVE pathway (*Le et al., 2021*). These roles, along with the previous implication of RAC1 signalling in early and later stages of PDAC (*Heid et al., 2011*; *Wu et al., 2014*), suggested potential involvement of CYRI in invasion and metastasis. Here, we demonstrate that CYRI-B is highly expressed in PDAC and can contribute to PDAC development, progression, and metastasis. We discover a role for CYRI-B in signalling that drives proliferation in early lesions. Later, during metastasis, we find that CYRI-B is required for chemotaxis towards LPA, implicating macropinocytic uptake of LPAR1 in PDAC metastasis. Our study highlights CYRI-B as a potentially interesting new target in PDAC progression and metastasis and further elucidates the molecular mechanisms underpinning metastatic spread.

## Results

### CYRI-B expression increases in precancerous lesions and PDAC

The *Cyrib* gene resides on human chromosome 8q24, near c-Myc, and is frequently amplified in many types of cancer, including pancreatic cancer (*Nikolaou and Machesky, 2020*). High expression of CYRI-B correlates with poor outcome in many cancers (*Li et al., 2021*; *Xu et al., 2022a*), including in human pancreatic cancer. To further investigate a potential role for *Cyrib* in pancreatic cancer, we first assessed the expression of CYRI-B in the KPC mouse model of metastatic PDAC (*Hingorani et al., 2005*). In this model, PanIN develops by around 10 weeks and progresses to later stages towards 15 weeks, with full-blown PDAC appearing at this stage and mice reaching end-point with a half time of median 150–200 days. As we do not currently have a reliable antibody to detect CYRI-B protein in tissue samples, tissue samples from 6-, 10-, 15-week-old KPC mice were processed for RNA in situ hybridisation (ISH). At 6 weeks, before appearance of PanIN, *Cyrib* was not detected in the pancreas (*Figure 1*). *Cyrib* expression was detectable by 10 weeks, especially around PanIN lesions, which remained stable until 15 weeks of age (*Figure 1*). End-point tumours showed a significant increase in the levels of *Cyrib* (*Figure 1*), suggesting that the KPC model was a good model for exploring the role of *Cyrib* expression during PDAC progression.

### CYRI-B deletion accelerates PDAC development, reducing the survival of mice

To further probe the mechanism by which CYRI-B might influence PDAC progression, we crossed *Cyrib* floxed mice with KPC mice. We refer to these mice and cell lines derived from them as CKPC (*Figure 2A*). ISH of end-point tumours confirmed no detectable *Cyrib* mRNA in CKPC tumours in comparison with KPC (*Figure 2B and C*). Western blotting also confirmed absence of CYRI-B protein in cell lines derived from end-point CKPC mouse tumours (CKPC-1 and CKPC-2) compared with a cell line from a KPC mouse tumour (KPC-1) (*Figure 2D*). KPC and CKPC end-point tumours showed no difference in the proliferation (Ki-67) or death (cleaved caspase-3, CC-3) of tumour cells (*Figure 2—figure supplement 1A–D*). CKPC tumours also did not show any significant change in the CD31 vessel density (*Figure 2—figure supplement 1E and F*) or necrosis (*Figure 2—figure supplement 1G and H*). However, there was a significant decrease in the median survival to end-point of CKPC mice (118 days) in comparison with the KPC mice (187 days) (*Figure 2E*) without affecting the tumour weight to body mass ratio at end-point (*Figure 2F*). Thus, loss of *Cyrib* in the pancreas accelerates progression to end-point of KRAS$^{G12D}$, p53$^{R172H}$-driven PDAC in the KPC model, but does not grossly alter levels of cell growth/death or histological appearance of end-point tumours.

### CYRI-B deletion accelerates PanIN formation

RAC1 is an important cancer driver downstream of KRAS and its ablation in mouse models delayed the onset of precancerous lesions (*Heid et al., 2011*) and led to an inability to sustain precancer progression to PDAC (*Wu et al., 2014*). Therefore, we asked whether loss of the RAC1 interactor

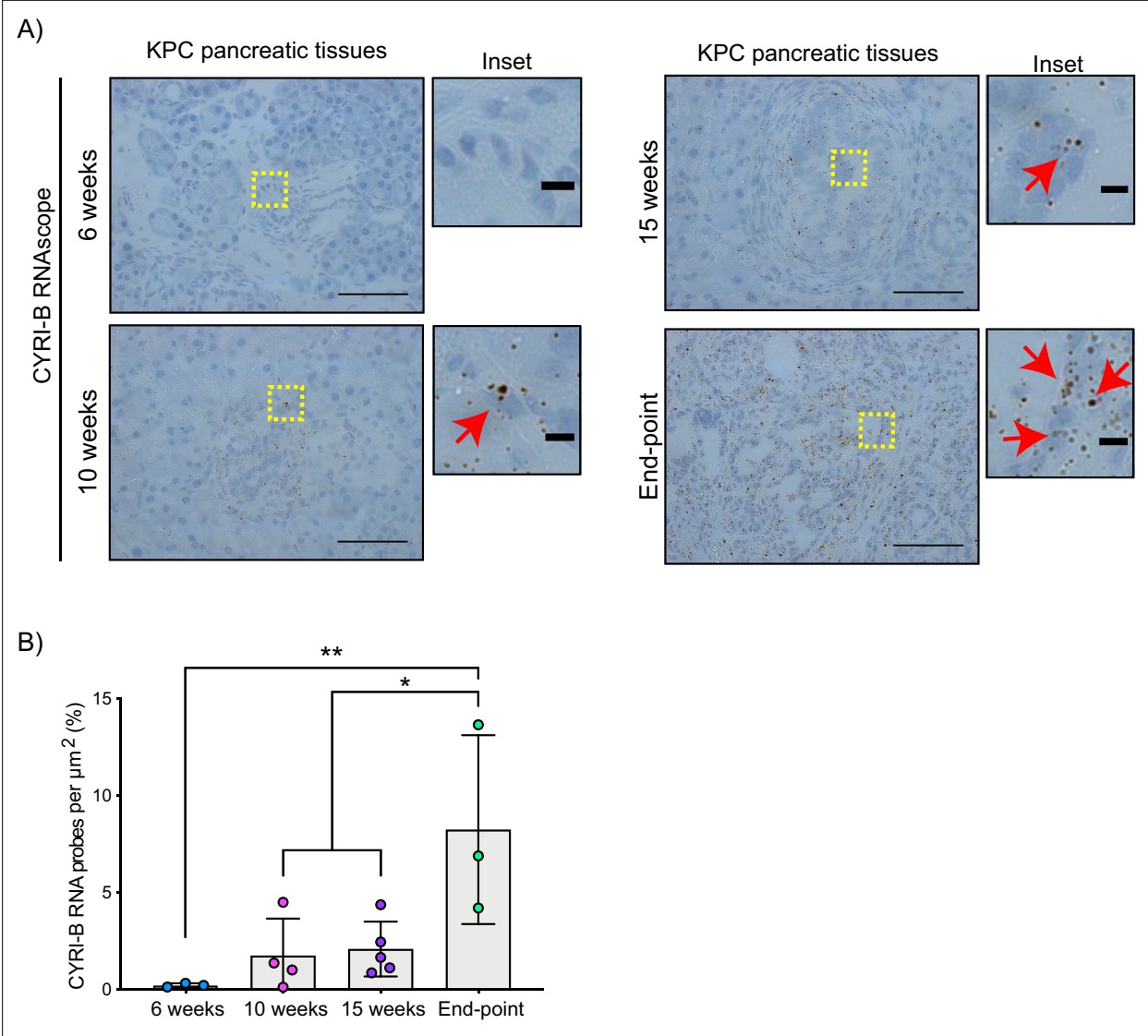

**Figure 1.** CYRI-B is expressed during pancreatic ductal adenocarcinoma (PDAC) progression. (**A**) Representative images of *Cyrib* RNAScope in situ hybridisation from 6-, 10-, 15-week-old and end-point KPC mouse tissues. RNA probes are visualised as brown dots. Haematoxylin was also used to stain the nuclei. Scale bars, 50 µm. Yellow boxes show the region of interest for magnified images (inset). Red arrows denote positive RNA probes. Scale bars, 5 µm. (**B**) Quantification of the CYRI-B RNA probes per µm$^2$ from (A). Mean ± SD; one-way ANOVA with Tukey's test was performed in n≥3 mice. *p<0.01, **p<0.001.

The online version of this article includes the following source data for figure 1:

**Source data 1.** *Cyrib* RNA probes detected per µm$^2$ in KPC mouse pancreatic tissues at 6, 10, 15 weeks and end-point tumours.

*Cyrib* affected the onset and progression between stages of PanIN1–3 (*Figure 3A and B*). Pancreatic samples from KPC or CKPC mice revealed no differences at 10 weeks, but more PanIN2 and -3 lesions were present in 15-week-old CKPC mice over KPC controls (*Figure 3C* and *Figure 3—figure supplement 1*), indicating an acceleration of early progression.

To further probe the role of *Cyrib* in lesion formation, we sought to understand the potential downstream signalling pathways that might be involved. RAC1 can drive cell proliferation through activation of both JNK and ERK downstream signalling pathways (*Bagrodia et al., 1995*; *Coso et al., 1995*; *Rul et al., 2002*; *Wang et al., 2019*). Therefore, we probed histological sections of pancreatic tissues from 15-week-old KPC and CKPC mice for pJNK and pERK. Consistent with enhanced RAC1 signalling, we observed a significant increase in the percentage pERK area and pJNK area from pancreata of CKPC mice vs KPC (*Figure 3D–G*). We next investigated proliferation using BrdU

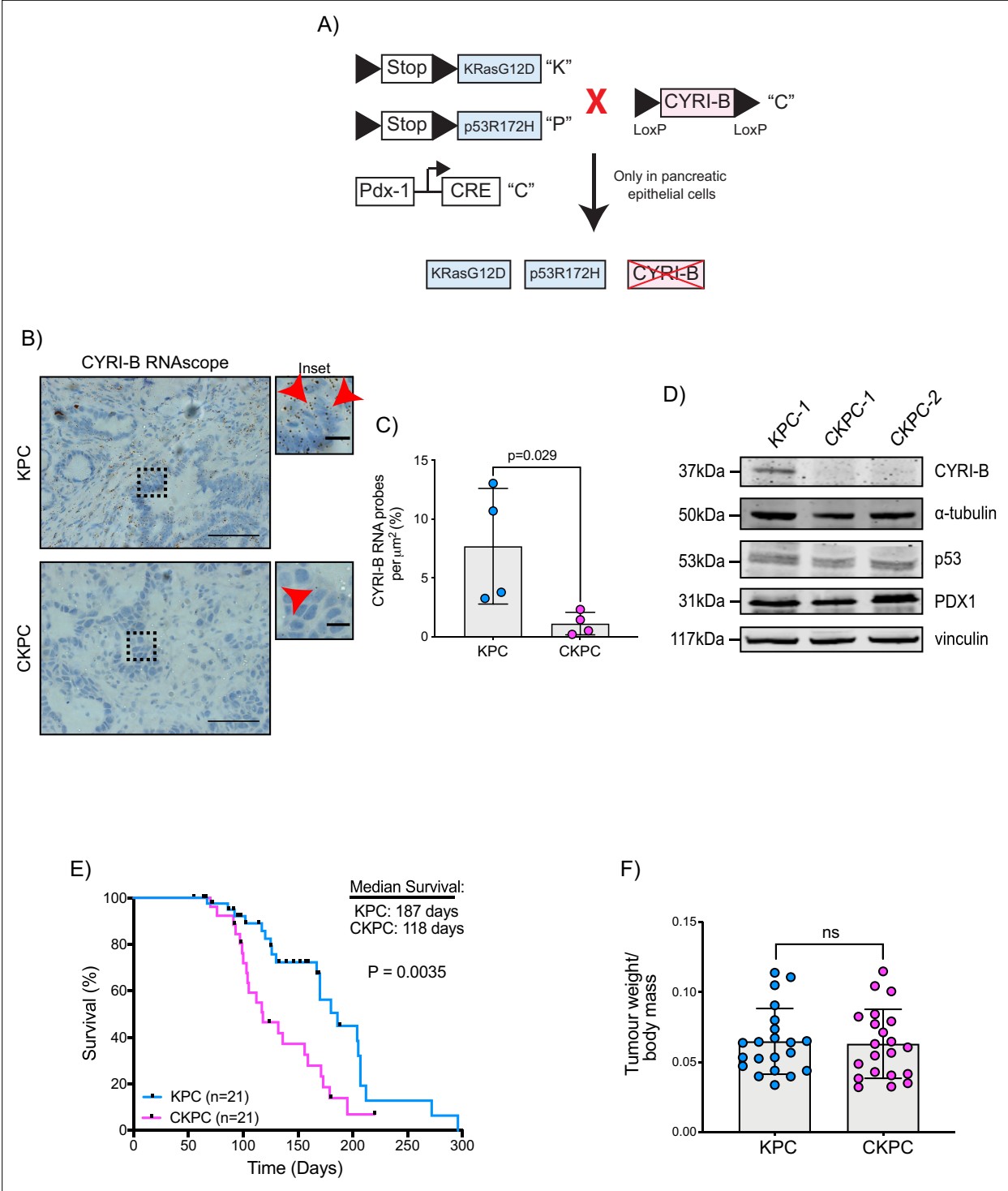

**Figure 2.** Loss of CYRI-B accelerates progression in the KPC mouse model of pancreatic ductal adenocarcinoma (PDAC). (**A**) Schematic representation of the CKPC mouse model. (**B**) Representative images for *Cyrib* RNAScope staining of end-point tumours from KPC and CKPC mice. Scale bars, 50 μm. Inset panels are magnified from the black dashed box. Scale bars, 10 μm. Red arrows indicate the positive *Cyrib* RNA. (**C**) Histograms showing the *Cyrib* RNA probes per μm² at end-point tumours in KPC and CKPC mice. Mean ± SD; unpaired t-test, n=4 KPC and 4 CKPC mice. (**D**) Representative western blot images of CYRI-B in cell lines established from one KPC (KPC-1) and two CKPC (CKPC-1 and CKPC-2) tumours. Membranes were also probed for anti-p53 and anti-PDX1 to validate the CKPC cells. α-Tubulin and vinculin were used as loading controls. Molecular weights as indicated on the side. (**E**) Survival (to end-point) curve (n=21 KPC, 21 CKPC independent mice). Log-rank (Mantel Cox) test used for comparing the KPC with CKPC survival

*Figure 2 continued on next page*

*Figure 2 continued*

curves. p-Value as indicated. (**F**) Histogram showing tumour to body mass ratios at sacrifice. Mean ± SD; unpaired t-test was performed in n=21 KPC and 21 CKPC mice. p-Value: not significant (ns).

The online version of this article includes the following source data and figure supplement(s) for figure 2:

**Source data 1.** *Cyrib* RNA probes detected per µm$^2$ in KPC mouse pancreatic tumours vs CKPC tumours at end-point.

**Source data 2.** Scans of original western blots unlabelled and labelled to support *Figure 2D*.

**Source data 3.** Data from Kaplan-Meier plot for survival of mice.

**Source data 4.** Tumour weight to body mass ratios for KPC and CKPC cohort mice.

**Figure supplement 1.** End-point CKPC tumours show comparable proliferation, apoptosis vascularisation, and necrosis to KPC tumours.

**Figure supplement 1—source data 1.** Percent area stained with Ki67+ nuclei in tumours from KPC vs CKPC mice.

**Figure supplement 1—source data 2.** Cleaved caspase 3 positive cells per area in tumours from KPC vs CKPC mice.

**Figure supplement 1—source data 3.** CD31 positive area in tumours from KPC vs CKPC mice.

**Figure supplement 1—source data 4.** Necrotic area in tumours from KPC vs CKPC mice.

injections at 15 weeks and found increased BrdU positive nuclei in the CKPC tissues in comparison with the KPC, suggesting enhanced proliferation in the abnormal ductal structures (*Figure 3H and I*). Indeed, CKPC mice presented with increased pancreatic weight to body mass ratio at 15 weeks, in agreement with increased proliferation of preneoplastic and neoplastic cells, whereas at 10 weeks of age there was no difference (*Figure 3J*). Thus, loss of *Cyrib* in the KPC model accelerates PanIN formation and progression, likely due to loss of CYRI-B's capacity to buffer RAC1 activation downstream of active KRAS leading to abnormal architecture, combined with hyperactivation of ERK and JNK to drive proliferation.

## CYRI-B regulates metastatic potential

The KPC mouse model is characterised by high metastatic rates to clinically relevant organs such as liver, diaphragm, and bowel (*Hingorani et al., 2005*). Since CYRI-B regulates cell migration and chemotaxis (*Fort et al., 2018*), we asked whether deletion of CYRI-B can affect the metastatic potential of cancer cells in the CKPC mouse model. Analysis of mice at end-point from KPC and CKPC cohorts revealed similar infrequent metastasis to the diaphragm in both cohorts, but a significant reduction in metastasis to both the liver and bowel of CKPC mice (*Figure 4A and B*).

To explore mechanisms behind the reduced metastasis of CKPC mice, we used an in vivo transplantation assay to test the metastatic seeding in the peritoneal cavity. This assay also allows us to rule out whether reduced metastasis was just due to the earlier progression to end-point in CKPC mice. *Cyrib* CRISPR (knockout of *cyri-b*, Ex 3) and control KPC-1 cells (*Figure 4—figure supplement 1A*), which show similar levels of proliferation (*Figure 4—figure supplement 1B*), were injected in the peritoneal cavity of nude mice and metastatic seeding was quantified. Although the pancreas weight to body mass ratio did not change (*Figure 4C*), there was a significant reduction in the formation of small metastatic buds on mesentery in the mice injected with *Cyrib* CRISPR KPC-1 cells (*Figure 4D and E*). No differences in proliferation were observed by Ki67 staining of tumours (*Figure 4F*). This mouse model also displays jaundice and ascites fluid, two symptoms which are very common in pancreatic cancer patients. There was a reduction in the number of mice with ascites fluid in mice bearing *Cyrib* CRISPR KPC-1 cells (*Figure 4G*). We did not observe any difference in the number of mice presenting with jaundice (*Figure 4H*), possibly because jaundice is caused by blockage of the bile duct, which could be a stochastic process, dependent on tumour position and other factors. In summary, CYRI-B is required for efficient metastatic seeding of KPC cells.

## CYRI-B deletion reduces chemotactic potential

Since we found that CYRI-B can influence the metastatic seeding of KPC cells, we sought to investigate whether CYRI-B can affect their chemotactic potential. Chemotaxis is a major driver of metastasis away from the primary tumour and towards sites rich in attractants, such as blood vessels. It was previously shown that the signalling lipid, LPA, which is found in blood serum, is an important chemoattractant driving melanoma and PDAC metastasis (*Juin et al., 2019*; *Muinonen-Martin et al., 2010*). LPA drives chemotaxis of KPC cells and can be sensed by the LPA receptor 1 (LPAR1), present

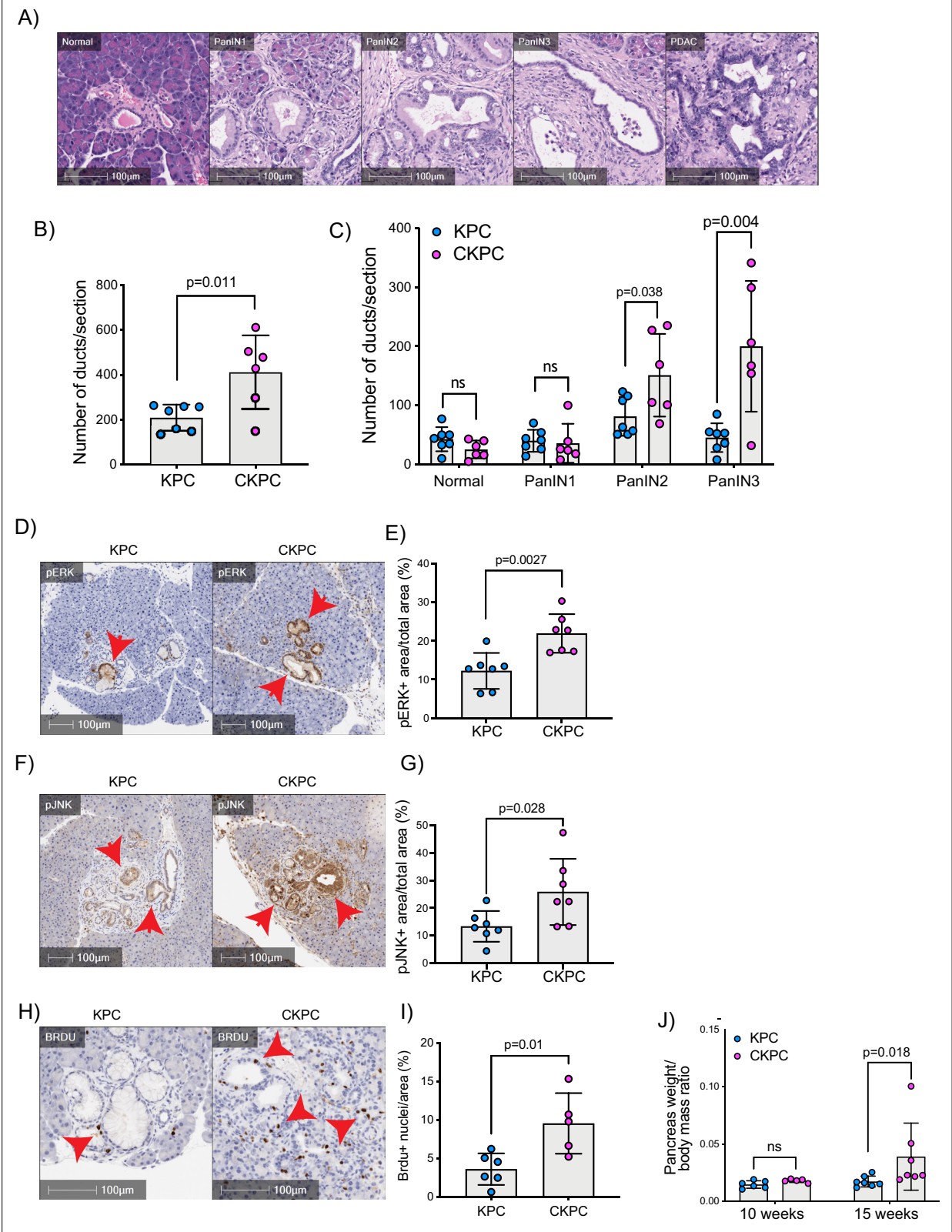

**Figure 3.** Loss of CYRI-B accelerates pancreatic intraepithelial neoplasm (PanIN) formation and increases pJNK, pERK, and proliferation. (**A**) Representative haematoxylin and eosin (H&E) images from KPC mice of normal pancreatic ducts, PanIN1, -2, -3 and pancreatic ductal adenocarcinoma (PDAC) lesions. Scale bars, 100 µm. (**B**) Number of ducts present in pancreas from 15-week-old KPC and CKPC mice (n≥6 mice). Mean ± SD; unpaired t-test was performed. p-Value as indicated. (**C**) Classification and scoring of pancreatic ducts in pancreas from 15-week-old KPC and

*Figure 3 continued on next page*

*Figure 3 continued*

CKPC mice (n≥6 mice). Mean ± SD; unpaired t-test was performed. ns = not significant, p-value as indicated. (**D**) Representative images of pancreata from 15-week-old mice stained with pERK and haematoxylin (nuclei). Red arrows indicate the positive pERK staining. Scale bars, 100 µm. (**E**) pERK positive area from the total quantified area from (D). Mean ± SD; unpaired t-test was performed in n=7 KPC and CKPC independent mice. p-Value as indicated. (**F**) Representative images of pancreata from 15-week-old-mice stained with pJNK and haematoxylin (nuclei). Red arrows indicate the positive pJNK staining. Scale bars, 100 µm. (**G**) pJNK positive area from the total quantified area from (**F**). Mean ± SD; unpaired t-test was performed in n=7 KPC and CKPC independent mice. p-Value as indicated. (**H**) Representative images of pancreatic tissue from 15-week-old KPC and CKPC mice stained for BrdU (proliferation) and haematoxylin. Red arrows show the BrdU positive nuclei. Scale bars, 100 µm. (**I**) Quantification of BrdU positive nuclei from KPC and CKPC 15-week-old pancreatic tissues. Mean ± SD; unpaired t-test was performed in n=6 KPC and 5 CKPC independent mice. p-Value as indicated. (**J**) Quantification of the pancreas to body mass ratio at 10 weeks (n=6 mice in each mouse model) and 15 weeks (n=7 in each mouse model) in KPC and CKPC mice. Mean ± SD; unpaired t-test was performed. ns = not significant, p-value as indicated.

The online version of this article includes the following source data and figure supplement(s) for figure 3:

**Source data 1.** Number of ductal structures per section for KPC and CKPC mice at 15 weeks.

**Source data 2.** Number of ductal structures per section for KPC and CKPC mice which have the grading of 'normal', pancreatic intraepithelial neoplasm (PanIN)1, PanIN2, PanIN3 at 15 weeks to support *Figure 3C*.

**Source data 3.** Percent area stained pERK+ in tumours from KPC vs CKPC mice to support *Figure 3E*.

**Source data 4.** Percent area stained pJNK+ in tumours from KPC vs CKPC mice.

**Source data 5.** Percentage area with BrdU+ nuclei in tumours from KPC vs CKPC mice.

**Source data 6.** Pancreas weight to body mass ratio in KPC and CKPC mice at 10 and 15 weeks.

**Figure supplement 1.** Loss of CYRI-B does not alter the formation of pancreatic intraepithelial neoplasm (PanIN) lesions in 10-week-old mice.

**Figure supplement 1—source data 1.** Number of ducts present in 10-week-old pancreas in KPC and CKPC mice.

at the plasma membrane of PDAC cells (*Juin et al., 2019*). We generated an independent CKPC cell line, CKPC-1, derived directly from CKPC tumours and rescued with CYRI-B-p17-GFP or GFP (*Figure 4—figure supplement 2A*). To confirm the phenotype, CKPC-1 GFP or rescued cells were seeded on fibronectin-coated glass and stained for ArpC2 to assess the localisation of the Arp2/3 complex at the leading edge (*Figure 4—figure supplement 2B*). CKPC-1 GFP cells presented with more lamellipodia, larger area, and increased ArpC2 recruitment to the leading edge in comparison with the rescued cells (*Figure 4—figure supplement 2B–E*) in line with previous results for other cell types (*Fort et al., 2018*).

Using Insall chemotaxis chambers (*Muinonen-Martin et al., 2010*), we investigated whether CKPC-1 cells can migrate up fetal bovine serum (FBS) gradients, which are a rich source of LPA. Both spider plots and rose plots showing the paths of individual cells and the mean resultant vector of migration, respectively, revealed that CKPC-1 cells (expressing GFP as a control) have dramatically reduced chemotactic potential towards FBS (*Figure 4I and J*). On the contrary, re-expressing GFP-tagged CYRI-B in CKPC-1 cells fully restored chemotaxis and directed migration towards FBS (*Figure 4I and J*). CKPC-1 rescued with GFP-tagged CYRI-B were also treated with LPA antagonist KI16425 (*Ohta et al., 2003*) showing that inhibition of LPAR1 and -3 by KI16425 abolished chemotactic steering, consistent with LPA being the major attractant in these conditions (*Figure 4I and J*). We have focussed on LPAR1 because we previously found that LPAR1 and not LPAR3 was the major chemotactic receptor expressed in PDAC cells (*Juin et al., 2019*). To further confirm that CYRI-B affects the chemotactic potential of PDAC cells, KPC control and *Cyrib* CRISPR cells were also assessed for their chemotactic ability towards serum (10% FBS) using Insall chambers. Deletion of *Cyrib* (Ex3 and Ex4) did not change the proliferation rate of cells (*Figure 4—figure supplement 1*), but reduced chemotactic migration in comparison with control cells (*Figure 4—figure supplement 3*). Therefore, CYRI-B is required for chemotactic migration towards serum LPA in PDAC cells.

## CYRI-B localises on macropinocytic cups and vesicles

Having shown that CYRI-B can influence the metastatic seeding of KPC tumours in vivo by regulating chemotactic migration, we further probed the role CYRI-B in chemotaxis. We first examined dynamic localisation of CYRI-B, using GFP-labelled CYRI-B (CYRI-B-p17-GFP) and live-cell imaging of both COS-7 cells and CKPC cells. Interestingly, CYRI-B was present on internal vesicles and tubules that connect with the vesicles (*Figure 5A* and *Figure 5—video 1*). The lifetime of vesicular CYRI-B containing structures was around 40s (*Figure 5B*), with an average diameter of about 1 µm (*Figure 5C*),

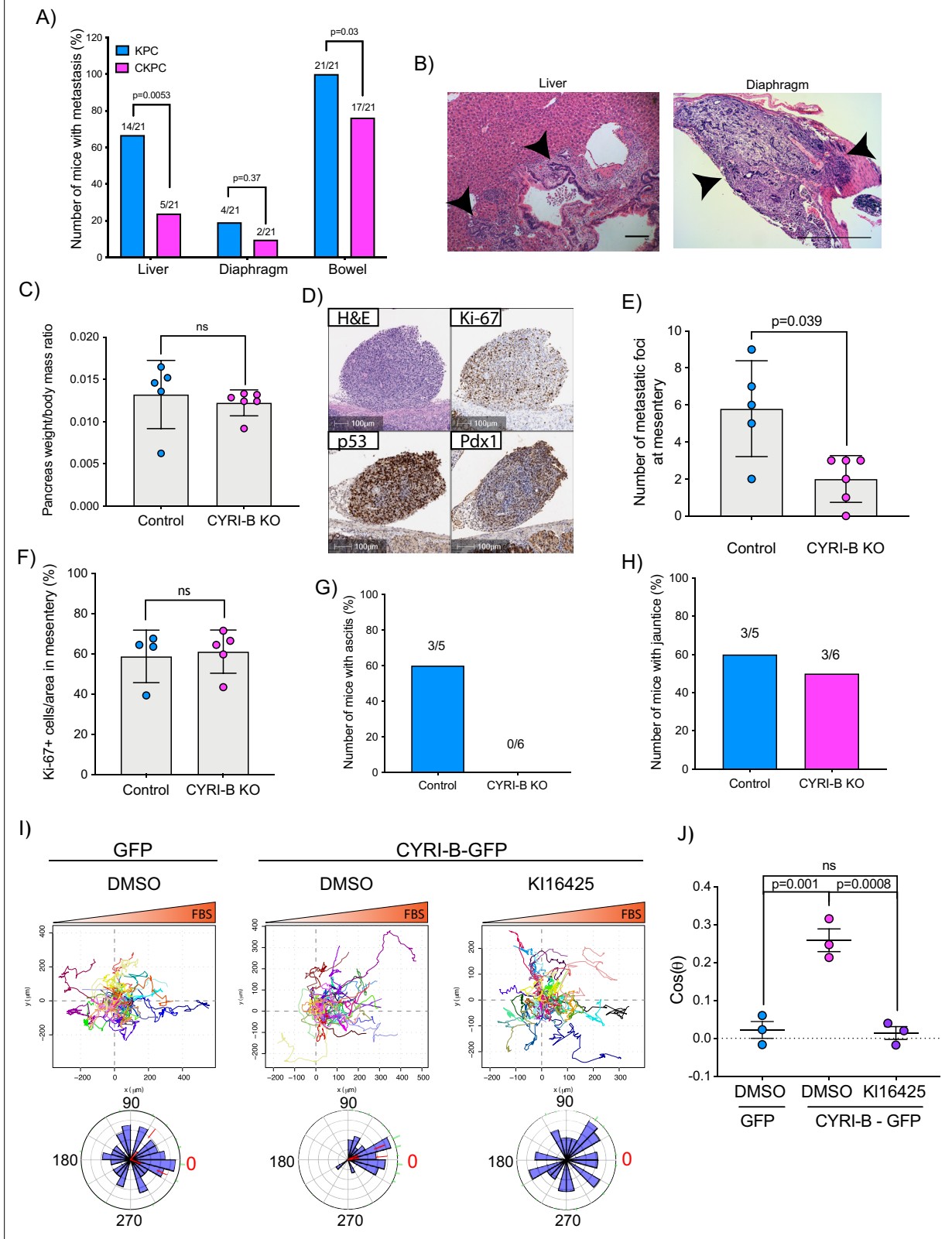

**Figure 4.** Loss of CYRI-B reduces metastasis and chemotactic potential. (**A**) Incidence of KPC or CKPC mice presenting with metastasis in liver, diaphragm, and bowel. Numbers above the bars show the fraction of mice with metastasis to the indicated site. Chi-square test was performed in n=21 KPC and CKPC mice. p-Value as indicated. (**B**) Representative haematoxylin and eosin (H&E) images of metastasis in the liver (scale bar, 50 μm) and diaphragm (scale bar, 100 μm). Black arrowheads denote metastatic lesions. (**C**) Histogram showing pancreas to body mass ratios at sacrifice.

*Figure 4 continued on next page*

*Figure 4 continued*

Mean ± SD; Mann-Whitney test was performed in n=5 for KPC control and n=6 mice for KPC *Cyrib* knockout (KO) cells. p-Value: not significant (ns). (**D**) Representative images of the mesenteric tumour foci from the in vivo transplantation assay. The metastatic foci were stained for H&E, Ki-67 (proliferation), p53, and PDX1 (for control). Scale bars, 100 µm. (**E**) Histogram of the number of metastatic foci at mesentery for KPC control and KPC *Cyrib* KO mice. Mean ± SD; Mann-Whitney test was performed in n≥5 mice for either control or *Cyrib* KO KPC injected cells. p-Value as indicated. (**F**) Quantification of the Ki-67 positive cells in the metastatic tumour foci. Mean ± SD; Mann-Whitney test was performed in n=4 for KPC control and n=5 mice for *Cyrib* KO KPC cells. p-Value: not significant (ns). (**G**) Incidence of mice presenting ascites (n≥5). (**H**) Incidence of mice presenting jaundice (n≥5). (**I**) Representative spider plots from n=3 independent chemotaxis assays of CKPC *Cyrib* KO and rescued cells. A chemotactic gradient of 10% foetal bovine serum (FBS) was established and cells were imaged for 16 hr (1 frame/15 min). Cells were also treated with either DMSO or the LPAR1/3 inhibitor KI16425 (10 mM) for 1 hr prior to imaging. Each cell trajectory is displayed with a different colour and the displacement of each cell is reported in the x- and y-axis. Orange gradient above shows the FBS gradient. Rose plot data are displayed for each condition below. Red dashed lines show the 95% confidence interval for the mean direction in the rose plots. The numbers represent degrees of the angle of migration relative to the chemoattractant gradient, with zero (red) denoting the direction of the chemoattractant gradient. (**J**) Quantification of the results in (**I**) showing the cos($\theta$) data (chemotactic index). Mean ± SEM from the average cos($\theta$) data of every repeat; one-way ANOVA followed by Tukey's multiple comparisons test was performed. p-Values as indicated on the graph, ns = not significant.

The online version of this article includes the following source data and figure supplement(s) for figure 4:

**Source data 1.** Spreadsheet with numerical data from *Figure 4C, E–H*.

**Source data 2.** Cos($\theta$) calculated for the chemotaxis assays shown in *Figure 4I*.

**Figure supplement 1.** Deletion of CYRI-B in KPC-1 cells does not affect proliferation.

**Figure supplement 1—source data 1.** Scans of original western blots unlabelled and labelled to support *Figure 4—figure supplement 1A*.

**Figure supplement 1—source data 2.** Growth curve displaying number of cells over time for control or EX3, EX4 CYRI knockout cells.

**Figure supplement 2.** Loss of CYRI-B results in enhanced spreading and Arp2/3 leading edge recruitment in pancreatic ductal adenocarcinoma (PDAC) cells.

**Figure supplement 2—source data 1.** Scans of original western blots unlabelled and labelled to support *Figure 4—figure supplement 2A*.

**Figure supplement 2—source data 2.** Number of cells presenting with lamellipodia or other protrusions in CYRI-B knockout (GFP) vs CYRI-B-GFP rescued (CYRI-B-GFP) cells.

**Figure supplement 2—source data 3.** Area per cell in CYRI-B knockout (GFP) vs CYRI-B-GFP rescued (CYRI-B-GFP) cells (*Figure 4*, *Figure 4—figure supplement 2C*).

**Figure supplement 2—source data 4.** Percentage of the periphery staining positive for ArpC2 in CYRI-B knockout (GFP) vs CYRI-B-GFP rescued (CYRI-B-GFP) cells.

**Figure supplement 2—source data 5.** Plasma membrane to cytoplasm relative intensity of ArpC2 in CYRI-B knockout (GFP) vs CYRI-B-GFP rescued (CYRI-B-GFP) cells.

**Figure supplement 3.** Deletion of CYRI-B abolishes chemotaxis.

**Figure supplement 3—source data 1.** Cos($\theta$) calculated from the chemotaxis assays shown in *Figure 4—figure supplement 3A*.

whereas the tubule length ranged up to 17 µm (*Figure 5D*). Additionally, we noticed CYRI-B localising at membrane cups (*Figure 5E* and *Figure 5—video 2*). CYRI-B positive pseudopods extend nascent cups, fuse together, and they slowly move inside the cells with a mean lifetime of about 19s (*Figure 5E*). Thus, CYRI-B localised on structures resembling macropinocytic cups, vesicles, and associated tubules, similar to what we previously described for CYRI-A (*Le et al., 2021*).

The CYRI-B positive cups and vesicles were in the size range of macropinosomes (0.2–5 µm), rather than other endocytic vesicles and cups, which are typically less than 0.2 µm (*Canton, 2018*). To test the role of CYRI-B in macropinocytosis, we added large molecular weight fluorescently labelled dextran (70 kDa) which can only enter by macropinocytosis (*Commisso et al., 2013*). It was important to test whether this occurred in PDAC cells, since previous work was done in other cell types (*Le et al., 2021*). CYRI-B stable CKPC-1 PDAC cell line ought to show enhanced macropinocytosis due to the active KRAS (*Commisso et al., 2013*; *Kamphorst et al., 2015*; *Palm et al., 2017*). In agreement with this, PDAC cells showed CYRI-B positive finger-like protrusions extending from the plasma membrane, until they fuse together engulfing extracellular dextran (*Figure 6A* and *Figure 6—video 1*). Live-cell imaging of COS-7 cells transfected with CYRI-B-p17-GFP also showed that CYRI-B positive pseudopods arising from the membrane enclose dextran, fuse together, and internalise (*Figure 6—figure supplement 1A* and *Figure 6—video 1*). The CYRI-B positive macropinosomes were then internalised and travelled inside the cells until they disappeared. Quantification of the lifetime of CYRI-B macropinosomes showed similar results between the two cell lines (*Figure 5B* and *Figure 6A*). Live-cell

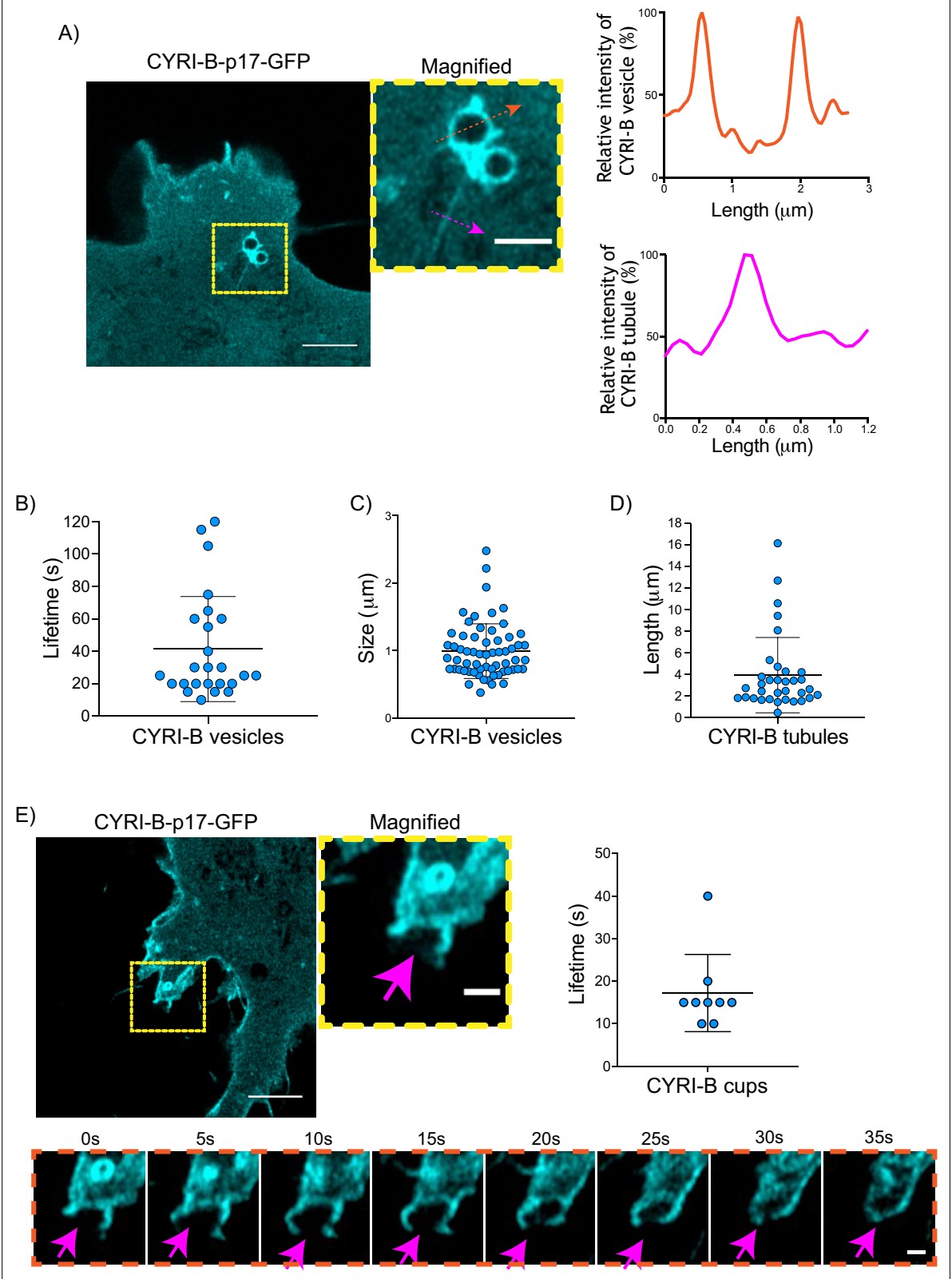

**Figure 5.** CYRI-B is localised at intracellular vesicles, tubules, and membrane cups. (**A**) Still image from live-cell videos of COS-7 *Cyrib* knockout (KO) cells transfected with CYRI-B-p17-GFP (cyan) - see *Figure 5—video 1*. Scale bar, 5 μm. Yellow box denotes magnified area. Magenta and orange arrows show the quantification area. Scale bar, 1 μm. Right panels show the quantifications of the relative intensity of the vesicles/cups and tubules. Image and quantification are representative of n=25 vesicles from a total of 10 cells, over 3 independent biological repeats. (**B**) Scatter plot of the lifetime of

*Figure 5 continued on next page*

Figure 5 continued

vesicles from (**A**). Error bars show the mean ± SD. (**C**) Scatter plot of the size (diameter) of CYRI-B positive vesicles from (A). Error bars show the mean ± SD. (**D**) Scatter plot of the length of CYRI-B tubules from (A). Error bars show the mean ± SD. (**E**) Still image from live-cell videos of COS-7 CYRI-B KO cells transfected with CYRI-B-p17-GFP (cyan), showing a macropinocytic cup - see *Figure 5—video 2*. Scale bar, 5 µm. Yellow box denotes magnified area. Magenta arrows show the quantification area. Scale bar, 1 µm. Scatter plot on the right panel shows the lifetime of the CYRI-B cups. Error bars show the mean ± SD. Orange dotted box shows the montage of the CYRI-B cup over time (**s**). Scale bar, 1 µm. Magenta arrows show the area of interest. Image and quantification are representative of n=9 events from a total of 4 cells.

The online version of this article includes the following video and source data for figure 5:

**Source data 1.** Lifetime in seconds of CYRI-B+ vesicles (*Figure 5B*).

**Source data 2.** Size in µm of CYRI-B+ vesicles (*Figure 5C*).

**Source data 3.** Length in µm of CYRI-B+ tubules (*Figure 5D*).

**Source data 4.** Lifetime in seconds of CYRI-B cups.

**Figure 5—video 1.** CYRI-B is localised at internal vesicles and tubules.

https://elifesciences.org/articles/83712/figures#fig5video1

**Figure 5—video 2.** CYRI-B is localised at membrane cups.

https://elifesciences.org/articles/83712/figures#fig5video2

imaging of AsPC-1 human pancreatic cancer cells transfected with CYRI-B-p17-GFP and mScarlet-Lck, a marker of the plasma membrane, showed that CYRI-B and Lck co-localised at the finger-like protrusions, cups, and internalised vesicles, confirming the localisation of CYRI-B on macropinosomes (*Figure 7B*, *Figure 6—figure supplement 1*, and *Figure 6—video 2*).

Thus, CYRI-B localises on macropinosomes in pancreatic cancer cell lines, suggesting a possible mechanism for how CYRI-B loss could affect tumour progression.

One of the first proteins to be recruited to macropinosomes, once they internalise, is Rab5, which is present on vesicles that move from the periphery of the cells towards the perinuclear region (*Bucci et al., 1994*; *Buckley and King, 2017*; *de Hoop et al., 1994*). Live-cell imaging of human AsPC-1 cells transfected with both CYRI-B-p17-GFP and Rab5a-mCherry showed that Rab5 is localised on macropinosomes (*Figure 7A*, *Figure 6—figure supplement 1*, and *Figure 7—video 1*). We found previously that CYRI-A showed a transient recruitment to macropinocytic cups and was largely absent from macropinosomes that had internalised, as marked by recruitment of the early endosome component Rab5 (*Puccini et al., 2022*). Therefore, we examined the localisation of CYRI-B relative to the early endosome component Rab5. Live-cell imaging of COS-7 cells transfected with both CYRI-B-p17-GFP and Rab5-mcherry showed that Rab5 is recruited after CYRI-B macropinosome internalisation. First CYRI-B positive pseudopods extend and fuse together to form the nascent macropinosome which is then internalised (*Figure 7B* and *Figure 7—video 2*). After ~50 s of internalisation, Rab5 is recruited to the macropinosomes (*Figure 7B* and *Figure 7—video 2*), suggesting that CYRI-B is present prior to and also during early macropinosome formation as marked with Rab5.

## LPAR1 internalises via CYRI-B positive macropinosomes

An important but often overlooked role of macropinocytosis is the maintenance of cell surface receptors (*Buckley and King, 2017*). Chemotaxis towards LPA requires the fine coordination of multiple proteins at the cell leading edge in order to sense LPA, internalise the LPAR1 receptor, and recycle it back to the plasma membrane (*Juin et al., 2019*; *Muinonen-Martin et al., 2010*). Having optimal amounts and dynamics of LPAR receptors at the leading edge is critical for a coordinated movement of cells towards the chemoattractant (*Juin et al., 2019*). In particular, we previously showed that LPAR1 is important for chemotaxis of pancreatic cancer cells (*Juin et al., 2019*). Therefore, we hypothesised that CYRI-B might influence the internalisation of LPAR1 at the leading edge. 70 kDa TRITC dextran was added to the medium and cells were imaged using live time lapse microscopy to visualise the macropinosomes in COS-7 cells transfected with LPAR-1 GFP. We observed that LPAR1-positive vesicles incorporated dextran and after some time they disappeared, with a mean lifetime of ~58 s (*Figure 8A*, *Figure 8—video 1*). Thus, LPAR1 is taken up from the cell surface by macropinocytosis.

To ask whether the LPAR1-positive macropinocytic structures also contained CYRI-B, we transfected CYRI-B-GFP stable CKPC-1 cells with LPAR1-mCherry and performed live-cell imaging. Indeed, upon CYRI-B internalisation, LPAR1 is also internalised with a lifetime of ~58 s, consistent with LPAR-1

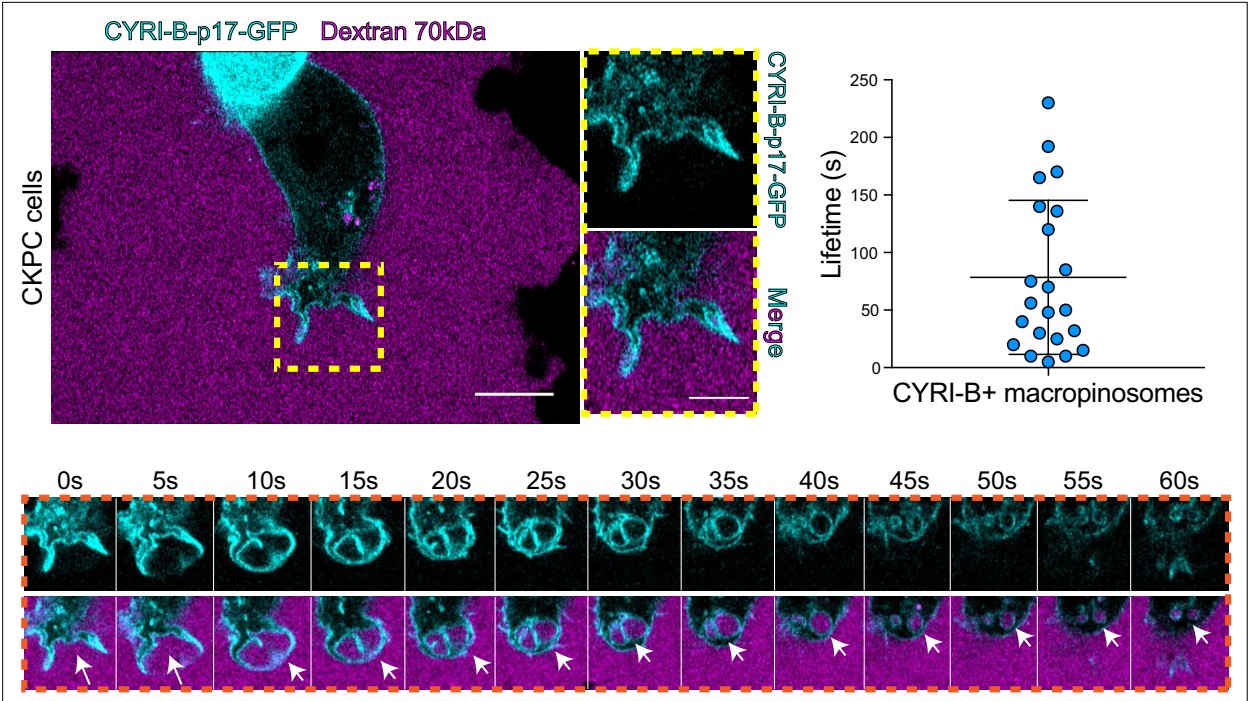

**Figure 6.** CYRI-B is recruited to macropinocytic cups. Still image from live-cell imaging of CKPC-CYRI-B-GFP stable cell lines (cyan) - see *Figure 6—video 1*. 70 kDa Dextran was added to the medium to visualise macropinocytic events (magenta). Scale bar, 10 µm. Yellow box shows the magnified area of interest, showing the macropinocytic cups. Scale bar, 5 µm. Scatter plot represents the lifetime of CYRI-B+ macropinosomes once internalised. Mean ± SD. Orange box shows a representative montage of CYRI-B internalisation via macropinocytosis. Scale bar, 1 µm. White arrows show CYRI-B localisation at the cups and the macropinosomes once internalised. n=21 events from a total of 6 cells.

The online version of this article includes the following video, source data, and figure supplement(s) for figure 6:

**Source data 1.** Lifetime in seconds of CYRI-B+ macropinosomes.

**Figure supplement 1.** CYRI-B localises at macropinocytic cups in COS-7 cells.

**Figure supplement 1—source data 1.** Lifetime in seconds of CYRI-B+ macropinosomes to support *Figure 6—figure supplement 1A*.

**Figure 6—video 1.** CYRI-B is localised at macropinocytic cups in COS-7 cells.
https://elifesciences.org/articles/83712/figures#fig6video1

**Figure 6—video 2.** CYRI-B is recruited to macropinocytic cups.
https://elifesciences.org/articles/83712/figures#fig6video2

---

trafficking via CYRI-B positive macropinocytic events in PDAC cells (*Figure 8B*, *Figure 8—video 2*). AsPC-1 cells transiently transfected with CYRI-B-GFP and LPAR1-mCherry also showed a co-localisation of CYRI-B-GFP and LPAR1 on macropinosomes (*Figure 6—figure supplement 1B*). Thus, LPAR1 is at least partially internalised via CYRI-B-mediated macropinocytosis. While it would be desirable to demonstrate the presence of endogenous LPAR-1 in macropinocytic cups, we are at present unable to localise endogenous LPAR1 with any available antibodies (see also *Juin et al., 2019*).

## CYRI-B controls chemotactic migration via macropinocytic LPAR-1 internalisation and membrane localisation

Having found that CYRI-B co-localises with LPAR1 and deletion of CYRI-B affects the chemotactic ability of PDAC cells to migrate in vitro and in vivo, we investigated whether CYRI-B could influence the trafficking of LPAR-1. Initial work showed that CKPC-1 cells expressing either GFP or CYRI-B-p17-GFP following stable transfection did not alter mRNA levels of LPAR1 or LPAR3 (*Figure 8—figure supplement 1A*), suggesting that changes in the chemotactic ability of the cells is not likely due to alterations in the expression level of the LPARs. Since CYRI-B alters the shape of CKPC cells, we also checked whether the localisation of LPAR is changed. CKPC-1 cells (with stable expression of GFP or CYRI-B-p17-GFP) were transfected with HA-LPAR1 and fixed for immunofluorescence. Cells

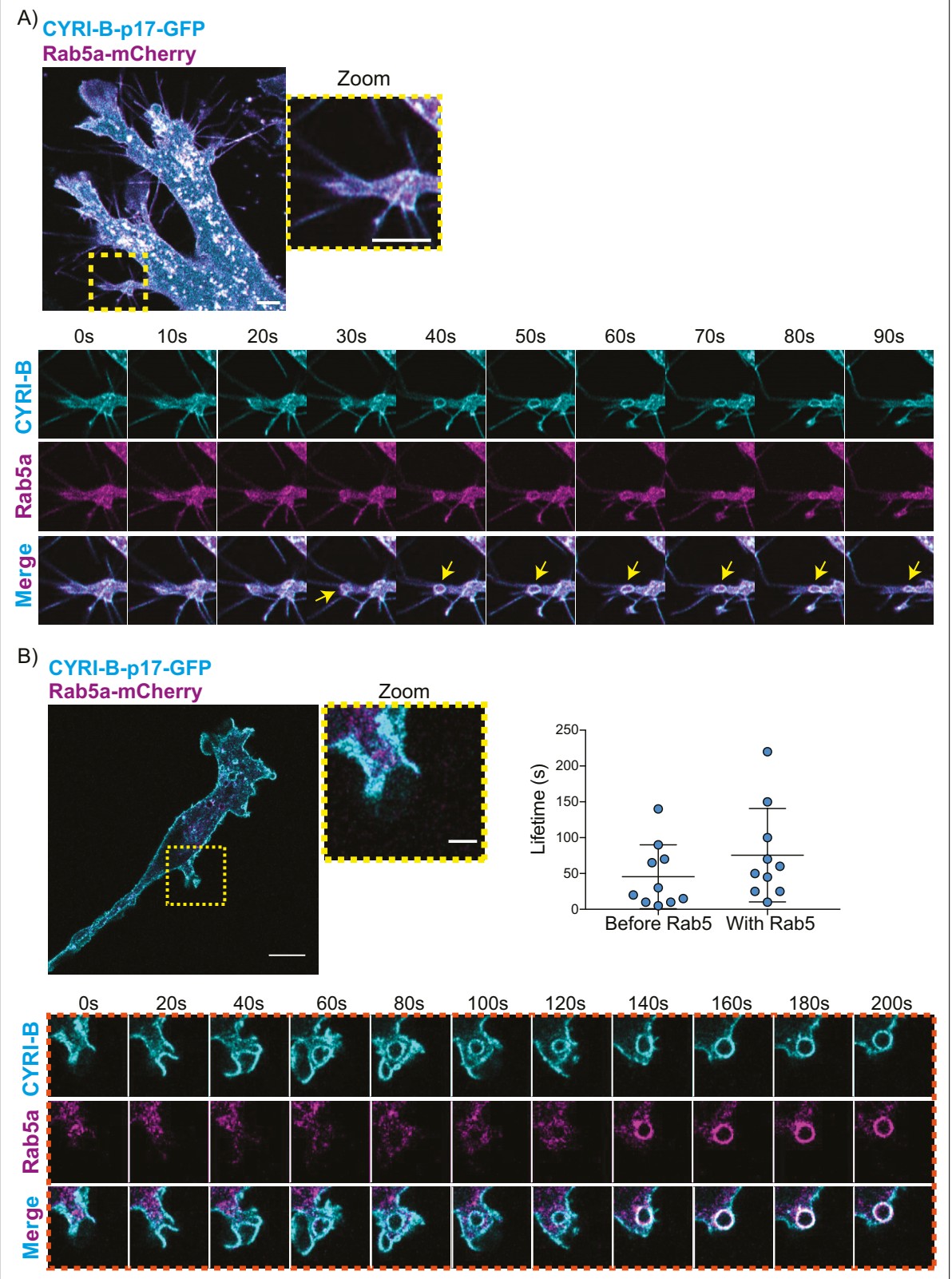

**Figure 7.** CYRI-B precedes Rab5 recruitment. (**A**) Still image from live-cell imaging of AsPC1 cells transiently transfected with CYRI-B-p17-GFP (cyan) and Rab5-mCherry (magenta) - see *Figure 7—video 1*. Scale bar, 5 μm. Yellow box shows the magnified area of interest, showing the macropinocytic cups. Scale bar, 5 μm. Yellow arrows show macropinosome. (**B**) Still image from live-cell imaging of COS-7 *Cyrib* knockout (KO) cells transfected with CYRI-B-p17-GFP (cyan) and mRFP-Rab5 (magenta) - see *Figure 7—video 2*. Scale bar, 10 μm. Yellow box show the magnified area of interest, showing

*Figure 7 continued on next page*

Figure 7 continued

the macropinocytic cups. Scale bar, 5 μm. Orange boxes show a representative montage of CYRI-B internalisation and the recruitment of Rab5 at the nascent macropinosomes. Scale bar, 5 μm. Scatter plots represent the lifetime of CYRI-B+ macropinosomes once internalised before and after Rab5 recruitment. Error bars show the mean ± SD; n=10 events from a total of 6 cells.

The online version of this article includes the following video and source data for figure 7:

Source data 1. Lifetime in seconds of CYRI-B+ macropinocytic structures before Rab5 arrival and after Rab5 arrival.

Figure 7—video 1. CYRI-B is localised at Rab5a positive macropinosomes in AsPC-1 cells.

https://elifesciences.org/articles/83712/figures#fig7video1

Figure 7—video 2. CYRI-B is recruited to macropinocytic cups and precedes Rab5 recruitment.

https://elifesciences.org/articles/83712/figures#fig7video2

displayed localisation of HA-LPAR1 at the plasma membrane and internal vesicles, as expected from previous reports (*Juin et al., 2019*). *Cyrib* knockout cells (expressing GFP-only) showed high levels of leading edge membrane localisation of LPAR1 in comparison with the CYRI-B-p17-GFP rescued cells (*Figure 8—figure supplement 1B and C*).

The combined evidence of the role of CYRI-B on macropinocytic uptake, the co-internalisation of CYRI-B with LPAR1, as well as the increase in peripherally accumulated LPAR1 led us to ask whether CYRI-B can affect the internalisation of LPAR1. We performed an image-based internalisation assay, using CKPC-1 stable GFP and CYRI-B-p17-GFP cells that were transfected with LPAR-1-mCherry and serum-starved overnight. Cells were stimulated with 10% FBS for 15 min and fixed to measure the internalisation of LPAR1. Previous reports suggested that stimulation of cells with serum should cause an increase in the internalisation of GPCRs including LPAR1 (*Juin et al., 2019*; *Kang et al., 2014*). CYRI-B rescued cells showed a serum-stimulated enhancement of LPAR1 internalisation, while CYRI-B depleted cells showed minimal stimulation of LPAR1 uptake (*Figure 9A and B*). While LPAR1 may be internalised by multiple endocytic pathways, the dependence on CYRI-B suggests that LPAR-1 is one cargo of CYRI-B-dependent macropinocytosis, perhaps together with integrin and other receptors (e.g. *Le et al., 2021*), regulating surface levels and chemotactic migration.

## Discussion

We have revealed an important role of CYRI-B in PDAC development, progression, and metastasis using the KPC mouse model and cells derived from the tumours. Our previous cell biology studies highlighted a role for CYRI-B as a buffer of RAC1-mediated actin assembly in lamellipodia and macropinocytic cups (*Fort et al., 2018*; *Le et al., 2021*), but did not address the potential role that CYRI-B could play in tumourigenesis and metastasis, given its central role as a regulator of motility.

We noticed that CYRI-B was highly expressed in human pancreatic cancers and correlated with poorer survival (*Nikolaou and Machesky, 2020*). Increased expression of CYRI-B in mice with PDAC suggested that this model could help to reveal the role of CYRI-B in human PDAC. We might predict, based on amplified expression of CYRI-B, that RAC1 activity might be dampened down at least during some stages of tumourigenesis. Having a buffer for RAC1 activity could provide advantages for tumours, where mutations in KRAS will drive high activation of RAC1, which might be detrimental to cell survival due to enhanced reactive oxygen species production and enhanced downstream signalling. While we were not able to measure RAC1 activity in tumours directly, it would be desirable to do this in the future, using such tools as a RAC1 FRET biosensor mouse model (*Floerchinger et al., 2021*). It would also be interesting to determine whether CYRI-A is involved in pancreatic cancer progression, as we previously implicated this protein in RAC1 binding (*Yelland et al., 2021*) and macropinocytosis (*Le et al., 2021*). However, to the best of our knowledge, there is not a correlation between CYRI-A expression and prognosis in human pancreatic tumours (e.g. Human Protein Atlas https://www.proteinatlas.org/ENSG00000197872-CYRIA/pathology). In contrast, CYRI-A appears to have prognostic potential in both renal and urothelial cancers, suggesting a possible tissue specificity (see Human Protein Atlas https://www.proteinatlas.org/ENSG00000197872-CYRIA/pathology).

Deletion of *Cyrib* in the pancreas, in concert with expression of KRAS $^{G12D}$ and p53 $^{R172H}$, led to acceleration of PanIN formation and an increase in the area of pancreas showing lesions with high phospho-ERK and phospho-JNK, two crucial downstream targets of KRAS and RAC1 that drive

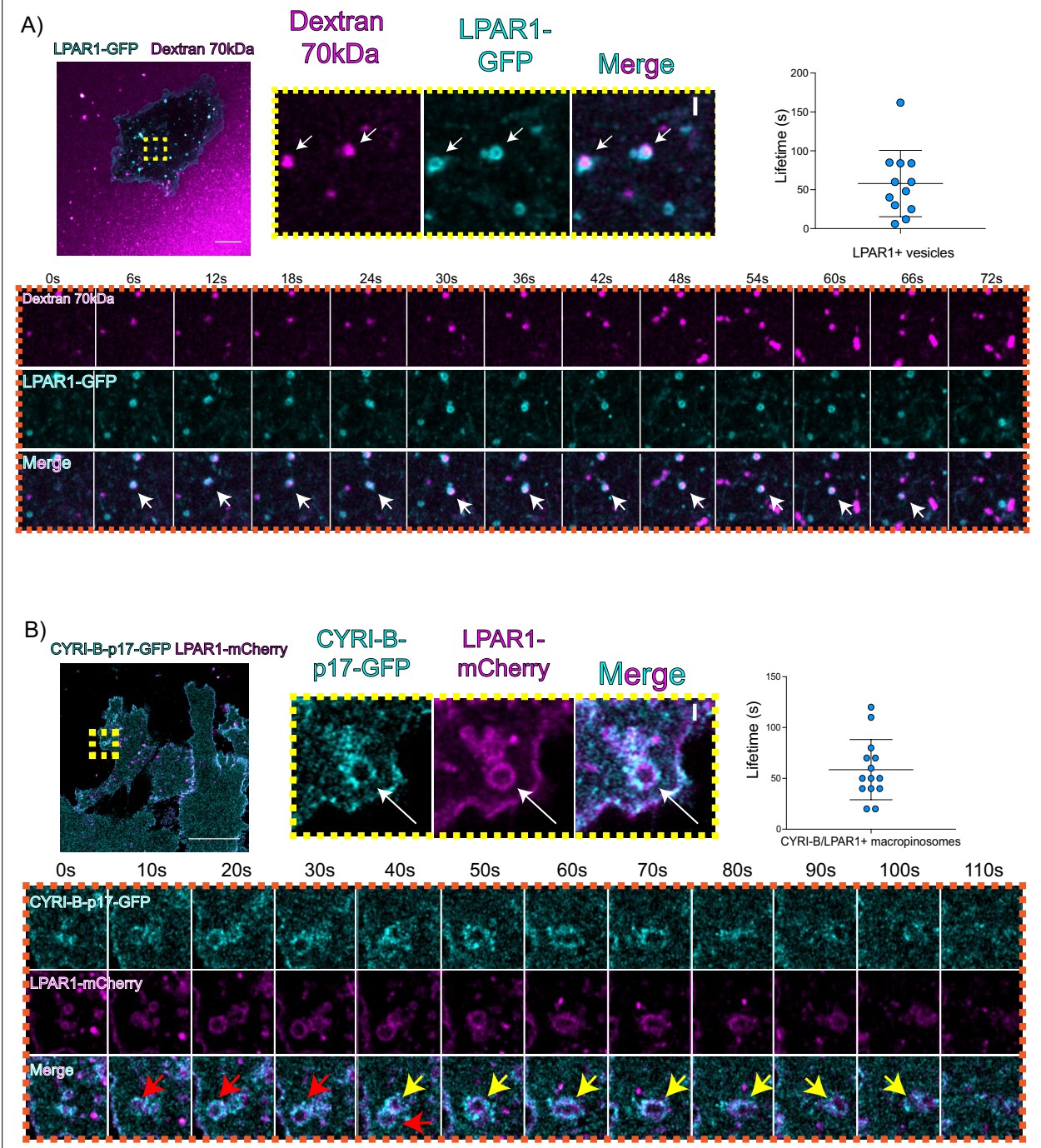

**Figure 8.** LPAR is internalised via CYRI-B positive macropinocytosis. (**A**) Still images from live-cell imaging of COS-7 cells transfected with LPAR1-GFP (cyan) - see *Figure 8—video 1*. 70 kDa Dextran was added to the medium to visualise the macropinosomes (magenta). Scale bar, 10 μm. Yellow box shows the magnified area of interest, showing the LPAR1+ macropinocytic vesicles/cups. White arrows denote structures of interest. Scale bar, 1 μm. Scatter plot represents the lifetime of LPAR1+ vesicles once internalised. Mean ± SD. Orange box shows a representative montage of LPAR1 internalisation via macropinocytosis. Scale bar, 1 μm. White arrows show the vesicle of interest. n=12 events from a total of 3 cells. (**B**) Still image from live-cell imaging of CKPC-1 cells transfected with CYRI-B-p17-GFP (cyan) and LPAR1-mCherry (magenta) - see *Figure 8—video 2*. Scale bar, 20 μm. Yellow box shows the magnified area of interest, showing the LPAR1 co-localisation with CYRI-B+ macropinosomes. White arrows show the vesicle of interest. Scale bar, 1 μm. Scatter plot represents the lifetime of LPAR1 and CYRI-B vesicles once internalised. Mean ± SD. Orange box shows a representative montage of LPAR1 and CYRI-B internalisation. Red and yellow arrows show the vesicles of interest. n=14 events from a total of 4 cells.

The online version of this article includes the following video, source data, and figure supplement(s) for figure 8:

**Source data 1.** Lifetime data for LPAR1+ vesicles.

*Figure 8 continued on next page*

*Figure 8 continued*

**Source data 2.** Lifetime data for LPAR1/CYRI-B+ vesicles.

**Figure supplement 1.** Loss of CYRI-B alters membrane localisation of LPAR1 but not its expression.

**Figure supplement 1—source data 1.** Fold-change of mRNA for LPAR1/LPAR1 normalised to GAPDH for CYRI-B knockout (GFP) and CYRI-B rescued (CYRI-B-GFP) cells.

**Figure supplement 1—source data 2.** Percent of cell periphery showing LPAR1+ staining in CYRI-B knockout (GFP) vs CYRI-B rescue (CYRI-B-GFP) cells.

**Figure 8—video 1.** LPAR1 internalises via macropinocytosis.

https://elifesciences.org/articles/83712/figures#fig8video1

**Figure 8—video 2.** LPAR1 internalises via CYRI-B positive macropinosomes.

https://elifesciences.org/articles/83712/figures#fig8video2

---

proliferation and expansion. These results support the idea that CYRI-B could buffer RAC1 activity in early tumourigenesis, but there are other possible explanations for its role. Another study implicated CYRI-B in mitochondrial superoxide production, which could fuel early tumour progression (*Chattaragada et al., 2018*). However, more studies are needed, as this is consistent with CYRI buffering RAC1, as RAC1 is a well-known regulator of superoxide production (*Du et al., 2011*).

Another possible mechanism by which loss of CYRI-B could enhance early cancer progression would be via its role in maintenance of epithelial apico-basolateral polarity (*Fort et al., 2018*). We previously found that depletion of CYRI-B in MDCK cell spheroids disrupted lumen formation in a similar way to hyperactivation of RAC1. Likewise, RAC1 and PI3-kinase are important for apicobasal polarity in the pancreas (*Löf-Öhlin et al., 2017*) and RAC1 has a known role in acinar to ductal metaplasia and in polarity and cell identity during early PDAC progression (*Heid et al., 2011*). PI3-kinase plays an important role in polarity maintenance and CYRI-B has been implicated in PI3-kinase signalling in gallbladder cancer cells (*Zhang et al., 2020*). Loss of CYRI-B could therefore lead to hyperactivation or inappropriate spatial control of RAC1 activation causing a loss of normal cell polarity and therefore enhancing preneoplasia and cancer progression. Polarity maintenance could be disrupted by a lack of proper control of Scar/WAVE complex localisation leading to aberrant actin regulation (*Fort et al., 2018*), or due to aberrant membrane trafficking of receptors such as integrins (*Le et al., 2021*) or LPAR1 (this study).

Other recent studies implicated CYRI-B as a potential biomarker for early cancer. CYRI-B autoantibodies were detected as a potential biomarker for early stage breast cancer (*Luo et al., 2022*), a gene found in patient serum on extrachromosomal circular DNA overexpressed in lung adenocarcinoma (*Xu et al., 2022b*) and a potential saliva marker of oral cancer (*Kawahara et al., 2016*). CYRI-B was also highlighted as a target of the zinc finger RNA-binding protein Zfrbp, leading to accelerated tumour development when overexpressed in colorectal and liver cancers (*Long et al., 2019*). Multiple studies also suggest that CYRI-B may be enriched in extracellular vesicles associated with cancer and other diseases (e.g. *Peng et al., 2019*). Taken together, CYRI-B may have potential as a biomarker and driver of early cancer and play a role in the progression or conversion from precancerous to cancerous lesions, but more studies are needed.

Involvement of CYRI-B in chemotactic migration suggested a mechanism for the reduced metastasis that we observed in both the KPC model and the intraperitoneal transplant model of PDAC metastasis. It is unclear why diaphragm and bowel metastasis were not significantly affected by CYRI-B deletion, but this might be due to the proximity of the tumours to these sites. In particular, pancreatic tumours appear to directly invade into the wall of the bowel due to the close proximity of these two organs and in human patients, duodenal invasion has also been observed (*Sopha et al., 2013*). We previously found that loss of N-WASP in the KPC model caused a reduction in metastasis and that the role of N-WASP in recycling the LPA receptor LPAR1 was crucial in mediating this phenotype (*Juin et al., 2019*). While N-WASP localises to SNX9-positive membrane tubules that are involved in trafficking of LPAR1 back to the plasma membrane after internalisation, we find that CYRI-B is required for efficient internalisation of LPAR1 after stimulation. Thus, CYRI-B and N-WASP control two different aspects of a similar pathway, whereby LPAR1 is stimulated, internalised, and then either sorted into tubules for rapid recycling or targeted towards lysosomes for slow recycling or degradation.

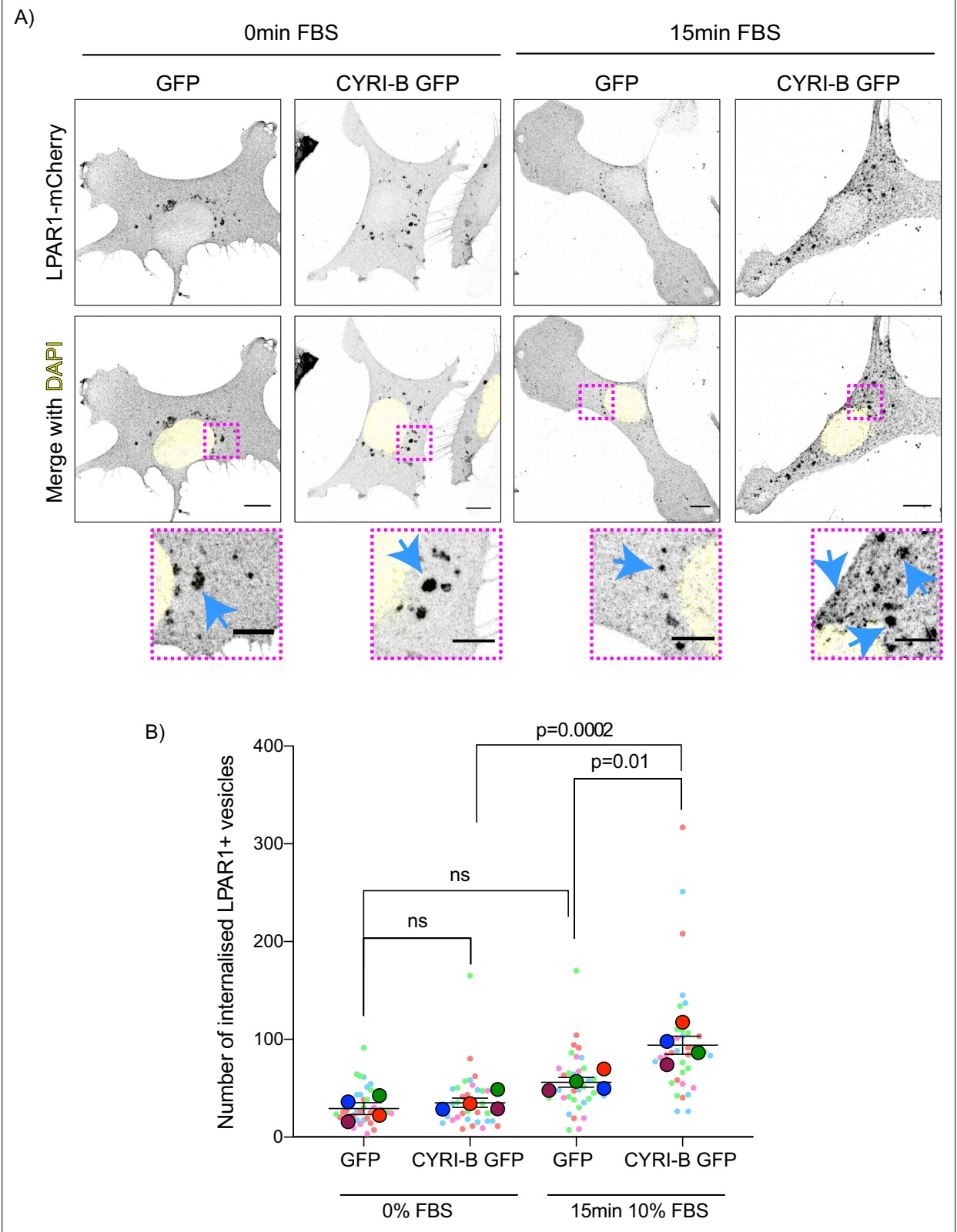

**Figure 9.** Loss of CYRI-B reduces LPAR1 internalisation upon serum stimulation. (**A**) Immunofluorescence images of CKPC-1 stable cells transfected with GFP or CYRI-B-p17-GFP. Cells were transfected with LPAR1-mCherry and seeded on fibronectin-coated coverslips. Cells were starved overnight and the next day 10% foetal bovine serum (FBS) was used to stimulate the uptake of LPAR1. Vesicles (marked by LPAR1-mCherry) are shown as black dots, DAPI (yellow) was used to visualise the nuclei. Scale bars, 10 µm. Magenta dotted boxes show the magnified area of interest and cyan arrows show the

*Figure 9 continued on next page*

*Figure 9 continued*
internalised vesicles. Scale bars, 5 μm. (**B**) Quantification of the number of LPAR1-positive vesicles in each condition. Scatter plot is presented as super plots and every independent biological repeat is coloured differently. Mean ± SEM; one-way ANOVA followed by Tukey's multiple comparisons test was performed, n=4 (from a total of ≥35 cells for each condition). p-Value as indicated, ns = not significant.

The online version of this article includes the following source data for figure 9:

**Source data 1.** Data for LPAR1+ vesicles as indicated with Excel and Prism data to support *Figure 9B*.

It may seem paradoxical that tumours overexpress CYRI-B, but are highly metastatic, as CYRI-B opposes RAC1 activity. However, we previously showed that CYRI-B is important for protrusion dynamics and that dynamics, e.g., assembly and rapid disassembly, is essential for chemotaxis (*Fort et al., 2018*). Cells with too much RAC1 activity and delocalised RAC1 activity do not chemotax effectively. They rather require a balance of activation and inactivation (*Pankov et al., 2005*). Similarly, with macropinocytosis, RAC1 needs to be transiently activated to allow actin assembly and cup formation, but then RAC1 is dampened and actin needs to be disassembled to allow closure and uptake of macropinosomes or phagocytic vesicles into the cell (*Le et al., 2021*; *Schlam et al., 2015*).

One limitation of our study was the lack of any antibodies to recognise endogenous CYRI-B or LPAR-1 in cells using immunohistochemistry or immunofluorescence. We acknowledge that this would be a major step forward, but so far, we have not found reliable antibodies for either of these targets.

Interestingly, a major mechanism for internalisation of LPAR1 appears to be via macropinocytosis, as we observed LPAR1 on the membrane surface of both nascent and internalised macropinocytic structures co-localising with CYRI-B. We also found a significant retardation of LPAR1 internalisation in CYRI-B depleted cells, indicating that LPAR1 is significantly internalised via CYRI-B-dependent macropinocytosis in PDAC cells. This suggests that macropinocytosis, which has recently attracted substantial interest as a regulator of nutrient uptake by PDAC cells (*Canton, 2018*; *Commisso et al., 2013*; *Michalopoulou et al., 2020*; *Puccini et al., 2022*; *Yao et al., 2019*), is also a key mechanism by which cells control surface receptor trafficking. Although involvement of macropinocytosis in receptor trafficking has been previously observed (reviewed in *Stow et al., 2020*), studies are primarily in immune cells, which perform high levels of constitutive uptake. Macropinocytosis as a way for cancer cells to control signalling and adhesion is perhaps under-appreciated and warrants further study in this capacity.

## Materials and methods

### Key resources table

| Reagent type (species) or resource | Designation | Source or reference | Identifiers | Additional information |
|---|---|---|---|---|
| Strain, strain background (*Mus musculus*) | Mouse: Pdx-1::Cre;KrasG12D; p53R172H (KPC) mice | *Hingorani et al., 2003* | | Can be obtained from CRUK Scotland Institute, Glasgow, UK |
| Strain, strain background (*Mus musculus*) | Mouse: Fam49b^fl/fl | This study | | Can be obtained from CRUK Scotland Institute, Glasgow, UK |
| Strain, strain background (*Mus musculus*) | CD-1 Nude Mice | Charles River | | |
| Cell line (*Cercopithecus aethiops*) | COS-7 cells | ATCC | CRL-1651 | |
| Cell line (*Homo sapiens*) | HEK293T | ATCC | CRL-3216 | |
| Cell line (*Homo sapiens*) | AsPC-1 | ATCC | CRL-1682 | |
| Transfected construct (human) | CYRI-B-p17-GFP | *Le et al., 2021* | | Can be obtained from Machesky Lab, Cambridge |

*Continued on next page*

*Continued*

| Reagent type (species) or resource | Designation | Source or reference | Identifiers | Additional information |
|---|---|---|---|---|
| Transfected construct (human) | pEGFP-N1 | Clontech- discontinued | | Can be obtained from Novopro, Catalogue number V012021 or Machesky Lab, Cambridge |
| Transfected construct (human) | LPAR1-mCherry | *Juin et al., 2019* | | Can be obtained from Machesky Lab, Cambridge |
| Transfected construct (human) | LPAR1-GFP | *Juin et al., 2019* | | Can be obtained from Machesky Lab, Cambridge |
| Transfected construct (human) | HA-LPAR1 | Kind gift from Dr. Heidi Welch | | Can be obtained from Machesky Lab, Cambridge |
| Transfected construct (human) | Rab5a-mCherry | Addgene | #55126, RRID Addgene 55126 | |
| Transfected construct (human) | mScarlet-Lck | *Le et al., 2021* | | Can be obtained from Machesky Lab, Cambridge |
| Transfected construct (human) | empty lentiCRISPRv1-puro | Addgene | #49535 | |
| Biological sample (*Mus musculus*) | Mouse KPC-1 cells (pdx-1::Cre;KrasG12D;p53R172) | Kind gift from Prof. J Morton | | Freshly isolated from KPC *Mus musculus*, see Materials and methods section |
| Biological sample (*Mus musculus*) | Mouse CKPC-1 and CKPC-2 PDAC cells (from BSNA9.4a male mouse) | This paper | | Freshly isolated from CKPC *Mus musculus*, see Materials and methods section |
| Biological sample (*Mus musculus*) | Mouse CKPC-2 PDAC cells (BSNA15.1a female mouse) | This paper | | Freshly isolated from CKPC *Mus musculus*, see Materials and methods section |
| Antibody | Anti-α-tubulin (DM1A) (Mouse monoclonal) | Sigma-Aldrich | CAT. #T6199 | WB: 1:1000 |
| Antibody | Anti-HA tag (C29F4) (Rabbit monoclonal) | Cell Signaling | CAT. #3724S | IF: 1:800 |
| Antibody | Anti-Fam49B (Rabbit polyclonal) | ProteinTech | CAT. #20127–1-AP | WB: 1:500 |
| Antibody | Anti-GAPDH (Mouse monoclonal) | Millipore | CAT. #MAB374 | WB: 1:2000 |
| Antibody | Anti-GFP (4B10) (Mouse monoclonal) | Cell Signaling | CAT. #2955 | WB: 1:1000 |
| Antibody | Anti-p53 antibody [PAb 240] (Mouse polyclonal) | Abcam | CAT. #Ab26 | WB: 2.5 µg/ml |
| Antibody | Anti-Pdx1 (D59H3) XP (Rabbit monoclonal) | Cell Signaling | CAT. #5679 | WB: 1:1000 |
| Antibody | Anti-ArpC2 (EPR8533) (Rabbit monoclonal) | Abcam | CAT. #133315 | IF: 1:500 |
| Antibody | Anti-BrdU (B44) (Mouse monoclonal) | BD Biosciences | CAT. #347580 RRID: AB_400326 | IHC: 1:250 |
| Antibody | Anti-Caspase 3 (ASP-175) (Rabbit polyclonal) | Cell Signaling | CAT. #9661 | IHC: 1:500 |
| Antibody | Anti-Ki67 (D3B5) (Rabbit monoclonal) | Cell Signaling | #12202 | IHC: 1:1000 |
| Antibody | Anti- p44/42 MAPK (ERK1/2) (Rabbit polyclonal) | Cell Signaling | #9101 | IHC: 1:400 |
| Antibody | Anti-pdx (Rabbit polyclonal) | Abcam | #ab47267 | IHC: 1:400 |
| Antibody | Anti-pSapk/Jnk Thr183/ Thr185 (81E11) (Rabbit Monoclonal) | Cell Signaling | #4668 | IHC: 1:20 |
| Antibody | Anti-Rabbit 680 nm stain (Donkey) | Invitrogen | CAT. #A21206 | IF: 1:10,000 |

*Continued on next page*

*Continued*

| Reagent type (species) or resource | Designation | Source or reference | Identifiers | Additional information |
|---|---|---|---|---|
| Antibody | Anti-Mouse 680 nm stain (Donkey) | Invitrogen | CAT. #A10038 | 1:10,000 |
| Antibody | Anti-Rabbit 594 nm (Donkey) | Invitrogen | CAT. #A21207 | 1:200 |
| Antibody | Anti-Mouse 800 nm (Goat) | Thermo Scientific | CAT. #SA5-35521 | 1:10,000 |
| Antibody | Anti-Rabbit 800 nm (Goat) | Thermo Scientific | CAT. #SA5-35571 | 1:200 |
| Antibody | Anti-Mouse 594 nm (Donkey) | Invitrogen | CAT. #A31203 | 1:200 |
| Sequence-based reagent | sgRNAs Mouse Cyri-b exon3 | This paper | DNA primer encoding sgRNA | CACCGGGTGCAGTCGTGCCACTAGT |
| Sequence-based reagent | sgRNAs Mouse Cyri-b exon4 | This paper | DNA primer encoding sgRNA | CACCGCGAGTATGGCGTACTAGTCA |
| Commercial assay or kit | Intense R Kit | Leica | DS9263 | |
| Commercial assay or kit | Rat ImmPRESS kit | Vector Labs | #MP-7404 | |
| Commercial assay or kit | RNAScope 2.5 LS (Brown) detection kit | Advanced Cell Diagnostics, Hayward, CA, USA | #322100 | |
| Commercial assay or kit | AMAXA-V kit | Lonza | VCA-1003 | |
| Commercial assay or kit | CRISPR-Cas9 calcium phosphate transfection | Invitrogen | CAT. #K2780-01 | |
| Commercial assay or kit | RNeasy Mini Kit | QIAGEN | CAT. #74104 | |
| Commercial assay or kit | DyNAmo HS SYBR Green qPCR kit | Thermo Fisher Scientific | CAT. #F410L | |
| Commercial assay or kit | Enzyme pre-treatment kit | Leica | CAT. #AR9551 | |
| Chemical compound, drug | KI16425 inhibitor | Cayman Chemicals | #10012659 | 1:1000 |
| Chemical compound, drug | Dextran, tetramethylrhodamine, 70,000 MW, lysine fixable (25MG) | Thermo Fisher | CAT. #D1818 | 50 µg/ml |
| Chemical compound, drug | Dextran, Fluorescein, 70,000 MW, Anionic, Lysine fixable | Thermo Fisher | CAT. #D1818 | 50 µg/ml |
| Chemical compound | Phalloidin 647 nm stain | Thermo Fisher | CAT. #A22287 | IF: 1:200 |
| Chemical compound | Phalloidin 594 nm stain | Thermo Fisher | CAT. #A12382 | IF: 1:200 |
| Chemical compound | Flex Wash buffer | Agilent | CAT. #K8007 | Use as per manufacturer's instructions |
| Chemical compound | High pH Target Retrieval Solution (TRS) | Agilent | CAT. #K8004 | Use as per manufacturer's instructions |
| Chemical compound | Liquid DAB | Agilent | CAT. #K3468 | Use as per manufacturer's instructions |
| Chemical compound | Low pH Target Retrieval Solution (TRS) | Agilent | CAT. #K8005 | Use as per manufacturer's instructions |
| Chemical compound | Mouse EnVision | Agilent | CAT. #4001 | Use as per manufacturer's instructions |

*Continued on next page*

*Continued*

| Reagent type (species) or resource | Designation | Source or reference | Identifiers | Additional information |
|---|---|---|---|---|
| Chemical compound | Peroxidase block | Agilent | CAT. #S2023 | Use as per manufacturer's instructions |
| Chemical compound | Rabbit EnVision | Agilent | CAT. #K4003 | Use as per manufacturer's instructions |
| Chemical compound | Rabbit signal boost (HRP) | Cell Signaling | CAT. #8114 | Use as per manufacturer's instructions |
| Chemical compound | Bond Wash | Leica | CAT. #AR9590 | Use as per manufacturer's instructions |
| Chemical compound | Epitope Retrieval Solution 1 (ER1) | Leica | CAT. #AR9551 | Use as per manufacturer's instructions |
| Chemical compound | Epitope Retrieval Solution 2 (ER2) | Leica | CAT. #AR9640 | Use as per manufacturer's instructions |
| Chemical compound | Mouse Ig Blocking reagent | Vector Labs | CAT. #MKB-2213 | Use as per manufacturer's instructions |
| Chemical compound | Fluoromount-G | Southern Biotech | CAT. #0100-01 | Use as per manufacturer's instructions |
| Software, algorithm | Fiji software | | RRID: SCR_002285 | |
| Software, algorithm | HALO software | Indica Labs | RRID:SCR_018350 | |
| Software, algorithm | Algorithm using R software | *Fort et al., 2018* | RRID:SCR_001905 | |

## Mouse model

The mice were maintained by the Biological Services Unit staff according to the UK home office regulations and instructions. The experiments were approved by the local Animal Welfare and Ethical Review Body (AWERB) of the University of Glasgow and performed under UK Home office licence PE494BE48 to LMM. Data for cohorts is included in *Supplementary file 1*. The genotyping was performed by ear notch and the samples were sent to TransnetYX. To generate the *Cyrib* floxed (Cyrib$^{fl/fl}$) mouse, frozen sperm was obtained from the Canadian Mouse Mutant Repository (Fam49b_tm1c_C08). The mouse strain was generated by IVF (*Takeo and Nakagata, 2011*; *Takeo and Nakagata, 2015*) using C57BL/6J mice as embryo donors, and the resulting two-cell embryos transferred to pseudopregnant recipients using standard protocols. The CKPC mouse model was generated by crossing *KRAS$^{LSL-G12D}$*, *Tp53$^{LSL-R172H}$*, *Pdx1-CRE* (KPC) mice (*Hingorani et al., 2003*) with *Cyrib* floxed (*Cyrib*$^{fl/fl}$) mice. Mice that died from causes other than pancreatic cancer were removed from the study.

## CKPC cell lines generation

CKPC cell lines (CKPC-1 and -2) were first generated by taking about 1/3 of the tumours from two different end-point mice. The tumours were washed three times with 5% penicillin/streptomycin (#15140122; Life Technologies) in PBS and cut into small pieces (<3 mm). The tumour pieces were then washed with PBS, centrifuged for 5 min at 1200 rpm, and transferred to 10 cm plates using full DMEM media supplemented with primocin (1:1000). The plates were left overnight in humidified incubator at 37°C supplied with 5% $CO_2$ until confluent. After about five to seven passages, cells were checked for Pdx-1, Tp53, and CYRI-B protein staining.

## Mammalian cell culture

COS-7 and HEK293T cell lines were cultured with Dulbecco's Modified Eagle's Medium (DMEM) (#21969-035; Gibco) growth medium supplemented with 10% FBS (#10270-106; Gibco) and 2 mM L-glutamine (#25030-032; Gibco). AsPC-1 cell line was cultured in RPMI medium 1640 (#31870-025; Gibco). COS-7 and HEK293T cell lines were split roughly every 2 days, AsPC-1 cells every 4 days, and maintained at 37°C humidified incubator and perfused with 5% $CO_2$.

For the proliferation assays, approximately $10^4$ KPC CRISPR or control cells were seeded on six-well plates and were manually counted every day for 4 days. Cells were tested regularly for mycoplasma and found to be negative. KPC cells were generated from pancreatic tumours of mice in our laboratory and verified by testing for expression of Pdx-1 pancreatic marker and TP53 (*Figure 2D*). Other cell lines (COS-7, HEK293T, and AsPC-1) were obtained from the ATCC https://www.atccc.org and were independently authenticated by short tandem repeat analysis.

## Cell transfection

About $1\times10^6$ COS-7 cells were transfected with Lipofectamine 2000 (#11668019, Invitrogen) according to the manufacturer's instructions. AsPC-1 cells were transfected with Lipofectamine 3000 (Invitrogen). Following the manufacturer's instructions, $2\times10^5$ cells were transfected with 5 µg of a combination of the following plasmids: Rab5a-mCherry or mScarlet-Lck. For KPC-1 and CKPC-1 cell lines, the AMAXA-V kit (VCA-1003, Lonza) was used according to the manufacturer's protocol. About $2\times10^6$ cells were electroporated using P-031 program from the AMAXA electroporator. The transfected cells were left overnight in full media in a humidified incubator at 37°C supplied with 5% $CO_2$.

For CYRI-B rescued stable cell line creation, cells were transfected with CYRI-B-p17-GFP along with a puromicin resistant plasmid using AMAXA-V kit according to the manufacturer's instructions. For control purposes the same CKPC cell line was also transfected with the empty GFP backbone. Cells were selected using 1 mg/ml puromicin and FACS sorted. Low-medium intensity GFP positive cells were selected to ensure that CYRI-B is not overexpressed. Cells were checked for CYRI-B expression and kept for maximum of three to four passages.

## sgRNAs and KPC CRISPR cell line generation

sgRNAs for CRISPR were designed using the Zhang laboratory website (https://zlab.bio/guide-design-resources). Mouse Cyri-b exon3 (CACCGGGTGCAGTCGTGCCACTAGT) and exon4 (CACCGCGAGTATGGCGTACTAGTCA) were used for CrispR cell line generation and transfected into lentiCRISPRv1-puro.

To generate *Cyrib* knockout stable cell lines, CRISPR-Cas9 genome editing technology was performed, using the calcium phosphate transfection kit (#K2780-01, Invitrogen). To generate the virus which infected the recipient cells (KPC or CKPC cells) the HEK293T cell line was used. First, about $2\times10^6$ HEK293T cells per 10 cm dish were seeded and let overnight to grow. Next day the transfection master mix which contained 10 µg of CRISPR construct containing sgRNA targeting the gene of interest (or empty lentiCRISPRv1-puro, #49535, Addgene), 7.5 µg of pSPAX2 (#12260, Addgene), and 4 µg of pVSVG packaging plasmid (#8454, Addgene) was prepared according to the manufacturer's instructions. The following day the medium was removed and replaced with the same medium composition (DMEM) with 20% FBS, for virus production. The cells were left overnight and in the meantime recipient cells were prepared for virus infection by seeding $1\times10^6$ cells per plate. The next day the medium from the HEK293T cells was removed and mixed with 2.5 µl hexadimethrine bromide (10 mg/ml) (#H9268, Sigma), filtered using a 0.45 µm pore membrane to remove any cell debris. The medium was then added to the recipient cells and left overnight. The next day the same procedure was repeated to achieve better infection with the virus. Transduced cells were selected using puromycin (2 µg/ml) (#ant-pr-1; InvivoGen).

## Chemotaxis assay

Chemotaxis assay was performed as previously described in *Muinonen-Martin et al., 2014*. About $2\times10^5$ cells were seeded on coverslips. Following attachment, the medium was replaced with SFM DMEM to starve the cells and left overnight. Next day the 'Insall' chemotaxis chambers were prepared. In the middle chamber, serum starvation medium was added. The coverslip was then carefully placed cell-side down onto the chamber. To create a chemoattractant gradient, full DMEM medium with 10% FBS was added on the sides of the Insall chambers (about 120 µl). The bridges containing the cells were then visualised every 15 min using a Nikon Ti long-term time-lapse microscope at 37°C with a ×10 objective for 48 hr. For LPAR1/3 inhibitor treatment with KI16425 antagonist (Cayman Chemicals, #10012659), cells were incubated for 1 hr in serum-free medium in 1:1000 dilution prior preparation and assembly of the 'Insall' chambers. DMSO (#15572393, Fisher Chemical) was used as a vehicle control. The cells were manually tracked using the mTrackJ plugin of Fiji software. From

each condition, at least two random bridges where selected and at least 25 cells from each bridge were manually tracked. Only cells present at the first frame of the video were counted in the tracking, whereas when the cells were moving outside the bridge the tracking was stopped. The Excel spreadsheets with all the cell tracks from each bridge were extracted in order to create rose plots, individual cell-track graphs, and cosθ data using an algorithm in R software which was previously designed and published in our lab (*Fort et al., 2018*).

## Western blotting

Protein extraction from cultured cells was performed using ice-cold RIPA (150 mM NaCl, 10 mM Tris-HCl pH 7.5, 1 mM EDTA, 1% Triton X-100, 0.1% SDS buffer) supplemented with 1× phosphatase and 1× protease inhibitors (#78427, #78438; Thermo Fisher Scientific). Cell lysates were centrifuged at 13,000 rpm at 4°C for 10 min and the supernatant was collected. Protein was quantified using the Precision Red (#ADV02; Cytoskeleton) advanced protein assay and 10–20 µg was used. The lysates were mixed with 1× NuPAGE LDS sample buffer (#NP0007, Invitrogen) and 1× NuPAGE reducing agent (#NP0004, Invitrogen), boiled for 5 min at 100°C and loaded on Novex 4–12% Bis-Tris acrylamide pre-cast gels (#NP0321; Thermo Fisher Scientific) at 170 V for about 1 hr. The proteins were transferred onto a 0.45 µm nitrocellulose blotting membrane (#10600002; GE Healthcare) using wet electrotransfer for 1 hr at 110 V. The membranes were blocked with 5% (wt/vol) BSA diluted in 1× TBS-T (10 mM Tris pH 8.0, 150 mM NaCl, 0.5% Tween-20) for 30 min at room temperature on a shaker. Primary antibodies were incubated in buffer with 5% (wt/vol) BSA and 1× TBS-T overnight at 4°C on a roller. Membranes were washed three times and incubated with Alexa Fluor-conjugated secondary antibodies (#A21206 and #A10038, Thermo Fisher Scientific) diluted in 5% (wt/vol) BSA and 1× TBS-T, for 1 hr at room temperature on a roller. The membranes were then washed three times and visualised using the Li-Cor Odyssey CLx scanner with the auto intensity scanning mode.

## Immunofluorescence assay

CKPC cells were plated onto sterile 13 mm glass coverslips that had been previously coated with 1 mg/ml fibronectin (#F1141; Sigma-Aldrich). Coverslips were fixed using 4% PFA (#15710, Electron Microscopy Sciences) in PBS for 10 min at room temperature. The coverslips were then washed three times with PBS and permeabilised with permeabilisation buffer for 5 min at room temperature. The coverslips were washed again three times with PBS and blocked with blocking buffer for about 30 min at room temperature. Primary antibodies were diluted in blocking buffer in the appropriate dilution and incubated for 1 hr at room temperature. Coverslips were washed three times with blocking buffer and secondary antibodies were then added in the appropriate dilution in blocking buffer and incubated for 1 hr at room temperature. Finally, the coverslips were washed three times with PBS and mounted on microscopy slides using Fluoromount-G solution containing DAPI (Southern Biotech; 0100-01). Slides were imaged using a Zeiss 880 LSM with Airyscan microscope.

## Macropinocytosis assays

Cells were seeded on fibronectin-coated coverslips and incubated for 2–4 hr. The culture medium was replaced with serum-free DMEM and left overnight in a 37°C in a humidified incubator perfused with 5% $CO_2$. The next day 10% FBS was added to the existing media and 0.2 mg/ml fluorescein-labelled dextran (70 kDa, D1822; Thermo Fisher Scientific) was added. The cells were either imaged live or incubated for 30 min, washed once with ice-cold PBS and immediately fixed with 4% PFA and stained for 30 min with DAPI. The coverslips were then washed thrice with PBS and mounted on microscopy slides.

## Live-cell imaging

For live-cell imaging cells were seeded on glass-bottom plates which were previously coated with either 1 mg/ml fibronectin (CKPC cells) or 10 µg/ml laminin (COS-7 cells, #L2020, Sigma). The cells were imaged using a Zeiss 880 LSM with Airyscan microscope which has a 37°C humidified incubator and perfused with 5% $CO_2$.

## Image-based LPAR1 internalisation assay

CKPC-1 stable cell lines were first transfected with LPAR1-mCherry construct. The cells were then seeded on fibronectin-coated coverslips and once adhered the media was replaced with serum-free DMEM overnight. The following day 10% FBS was added to the pre-existing medium and incubated for 15 min for the internalisation to occur. The media was aspirated and immediately fixed using 4% PFA. The cells were then mounted on microscopy slides and visualised using a Zeiss 880 LSM with Airyscan microscope (×63, oil, 1.4NA objective). Images were analysed by thresholding the LPAR1-mCherry channel and analysing the objects which were 0.1 µm or above in Fiji software.

## Reverse transcription quantitative polymerase chain reaction

RNA was first isolated from the cells using the RNeasy Mini Kit (#74104, QIAGEN) according to the manufacturer's instructions. The cDNA was then synthesised using the SuperScript III Reverse Transcriptase protocol and measured using NanoDrop2000c. 1 µg of cDNA was mixed along with a primer master mix previously made. The quantitative polymerase chain reaction (qPCR) was performed according to the manufacturer's instructions and the DyNAmo HS SYBR Green qPCR kit (#F410L; Thermo Fisher Scientific) reaction was set up as follows using the C1000 Thermal Cycler (CFX96 Real time system, Bio-Rad): 3 min at 95°C, 20 s at 95°C, 30 s at 57°C, 30 s at 72°C, repeat steps 2–4 for 40 cycles and 5 min at 72°C. Each condition had three technical replicates and GAPDH was used as a housekeeping gene (Fw - CATGGCCTACATGGCCTCCA, Rv - TGGGATAGGGCCTCTCTTGC, Thermo Fisher). The mRNA fold change was calculated using the $\Delta\Delta C_t$ method. The LPAR1 and LPAR3 Quantitec primers were purchased from QIAGEN (#QT00264320, #QT00107709).

## In vivo transplantation assay

For the in vivo transplantation assay KPC-1 CYRI-B CRISPR cell lines were used. The cells were grown in full media as normal until 24 hr before transplant, when fresh media was added without any antibiotics. About $2\times10^6$ cells per mouse were used for the experiment. Cells were injected into the intraperitoneal cavity of 10-week-old female CD-1 mice (Charles Rivers). Once injected, the mice were monitored every day and at day 14 mice were sacrificed. The weight of the mice and pancreata were taken at end-point.

## Immunohistochemistry, in situ hybridisation detection (RNAScope) assays, and quantification

Tissues were fixed in 10% formalin and next day transferred to 70% ethanol. The tissues were then embedded into paraffin blocks. For immunohistochemistry, the staining was performed on 4 µm sections which had previously been ovened at 60°C for 2 hr, using standard protocols. The detection for mouse *Cyrib* mRNA was performed using RNAScope 2.5 LS (Brown) detection kit (#322100; Advanced Cell Diagnostics, Hayward, CA, USA) on a Leica Bond Rx autostainer strictly according to the manufacturer's instructions. Slides were imaged using the Leica SCN 400f scanner.

To quantify the histology slides, the HALO software was used. About eight different areas (>350,000 µm² each area) within the pancreatic tumours were used to quantify the different stains. Necrotic areas were quantified manually from the whole pancreatic tumour using haematoxylin and eosin (H&E) staining. Areas with fragmented nuclei were considered as positive for necrosis. For tissues from 15-week-old mice, both neoplastic lesions and tumour areas were quantified (pJNK, pERK, BrdU stains). For the RNAScope experiments, the algorithm was set up to recognise the individual dots. PanIN lesion quantification was performed manually according to the following website: https://pathology.jhu.edu/pancreas/medical-professionals/duct-lesions using H&E slides.

## Statistical analysis

Data and statistics were analysed using the Prism software. All of the cell biology experiments, unless otherwise stated in the figure legends, were performed three times on separate occasions with separate cell passages. For all the mouse experiments and histology quantifications for end-point mice between KPC and CKPC mouse models, similar ages of mice were chosen. Unless otherwise stated, all the cell biology experiments were plotted as super plots (*Lord et al., 2020*) and each biological replicate was coloured differently. The quantified numbers of individual cells from the repeats are shown as individual points in the background.

Technical replicates are shown as smaller shapes and coloured depending on the repeat. Generated graphs show either SEM or SD depending on the experiment. ns=not significant/$p>0.05$, *$p<0.05$, **$p\leq0.01$, ***$p\leq0.001$, ****$p\leq0.0001$, unless the exact p-value is shown on the graph.

## Acknowledgements

We thank Cancer Research UK for core funding (A17196 and A31287) and funding to LMM (A24452 and DRCPG100017) and UKRI EPSRC grant (EP/T002123/1) to LMM. We thank the BAIR imaging facility for assistance with microscopy experiments, Dr Colin Nixon and the Histology facility for help with the tissue histology, the Beatson Biological Services Unit for their help with breeding, maintenance, and animal experiments.

## Additional information

### Funding

| Funder | Grant reference number | Author |
| --- | --- | --- |
| Cancer Research UK | A24452 | Laura M Machesky |
| Cancer Research UK | A17196 | Amelie Juin<br>Jamie A Whitelaw<br>Nikki R Paul<br>Loic Fort<br>Colin Nixon<br>Heather J Spence<br>Sheila Bryson<br>Laura M Machesky |
| Cancer Research UK | A31287 | Amelie Juin<br>Jamie A Whitelaw<br>Nikki R Paul<br>Loic Fort<br>Colin Nixon<br>Heather J Spence<br>Sheila Bryson<br>Laura M Machesky |
| EPSRC UKRI | EP/T00213/1 | Laura M Machesky |
| Cancer Research UK | DRCPG100017 | Laura M Machesky |

The funders had no role in study design, data collection and interpretation, or the decision to submit the work for publication.

### Author contributions

Savvas Nikolaou, Conceptualization, Formal analysis, Investigation, Methodology, Writing – original draft, Writing – review and editing; Amelie Juin, Conceptualization, Formal analysis, Investigation, Methodology, Writing – review and editing; Jamie A Whitelaw, Conceptualization, Investigation, Writing – review and editing; Nikki R Paul, Heather J Spence, Investigation, Methodology; Loic Fort, Resources, Methodology, Writing – review and editing; Colin Nixon, Methodology, Writing – review and editing; Sheila Bryson, Resources, Methodology; Laura M Machesky, Conceptualization, Supervision, Funding acquisition, Writing – original draft, Project administration, Writing – review and editing

### Author ORCIDs

Savvas Nikolaou (ID) https://orcid.org/0000-0001-6936-5911
Amelie Juin (ID) https://orcid.org/0000-0002-1481-000X
Jamie A Whitelaw (ID) https://orcid.org/0000-0001-6739-1032
Loic Fort (ID) https://orcid.org/0000-0001-6939-3621
Colin Nixon (ID) https://orcid.org/0000-0002-8085-2160
Laura M Machesky (ID) https://orcid.org/0000-0002-7592-9856

## Ethics

Mice were maintained by the Biological Services Unit staff according to the UK home office regulations and instructions. The experiments were approved by the local Animal Welfare and Ethical Review Body (AWERB) of the University of Glasgow and performed under UK Home office licence PE494BE48 to LMM.

## Decision letter and Author response

Decision letter https://doi.org/10.7554/eLife.83712.sa1
Author response https://doi.org/10.7554/eLife.83712.sa2

# Additional files

## Supplementary files
• Supplementary file 1. Spreadsheet with data on cohort animals. Table outlining data on mouse cohorts used in the experiments.
• MDAR checklist

## Data availability

All Western blot and numerical data generated or analysed during this study are included in the manuscript and supporting files; source data files have been provided for all graphs and western blots. The original data for this paper has been deposited to the University of Glasgow's Enlighten Database with the DOI https://doi.org/10.5525/gla.researchdata.1371.

The following dataset was generated:

| Author(s) | Year | Dataset title | Dataset URL | Database and Identifier |
| --- | --- | --- | --- | --- |
| Nikolaou S, Juin A, Whitelaw J, Paul N, Fort L, Nixon C, Spence H, Bryson S, Machesky L | 2022 | CYRI-B mediated macropinocytosis drives metastasis via lysophosphatidic acid receptor uptake | https://doi.org/10.5525/gla.researchdata.1371 | University of Glasgow, 10.5525/gla.researchdata.1371 |

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
