## [Editor Report]

This important study combines in vivo and in vitro models to characterise the complex role of CYRI-B, an interactor of the small GTPase Rac1, in controlling pancreatic cancer progression towards a higher proliferative and metastatic stage. The authors demonstrate that CYRI-B reduces the typical hyperactivation of Rac1 in early stages of tumor progression; subsequently, CYRI-B mediates internalization of lysophosphatidic acid receptor 1 (LPAR1) uptake through macropinocytosis, thus regulating chemotactic migration of cancer cells towards lysophosphatidic acid (LPA). This work, based on convincing evidence, will be of broad interest to cell biologists and the signalling research communities.

---

## [Decision Letter]

**Decision letter after peer review:**

Thank you for submitting your article "CYRI-B mediated macropinocytosis drives metastasis via lysophosphatidic acid receptor uptake" for consideration by *eLife*. Your article has been reviewed by 3 peer reviewers, and the evaluation has been overseen by a Reviewing Editor and Erica Golemis as the Senior Editor. The following individuals involved in the review of your submission have agreed to reveal their identity: Vania MM Braga (Reviewer #2); Stefan Linder (Reviewer #3).

Essential revisions (for the authors):

The manuscript has been improved but there are some remaining issues that need to be addressed, as outlined below. Besides these comments, a number of suggestions have been made by the reviewers, which appear following the public reviews; the authors are encouraged to address as many of these as possible:

1. The authors argue that CYRI-B could act as a buffer for Rac1 hyperactivation. This is likely based on earlier work of the authors using MDCK cells. However, no data on Rac1 activation and its regulation by CYRI-B in PDAC are shown. Please include respective data or discuss the possible, also alternative, molecular roles for CYRI-B in PDAC progression in more detail.

2. It is not clear why the authors use COS cells for showing the role of CYRI-B in macropinocytosis, as also some data using CKPC cells are shown, which should be more relevant. For example, Figure 7B is the only figure showing CYRI-B-GFP and LPAR-1-mCherry on macropinosomes. However, this gallery does not show the beginning of the process. It would be important to show in CKPC cells that both proteins are already present at nascent macropinocytic cups that then lead to fully formed macropinosomes that are also positive for Rab5. So far, these results are a mosaic of data from COS and CKPC cells.

3. Related to the previous point: it would also be important to detect endogenous CYRI-B and LPAR1 in PDAC cells and to show their localization at macropinocytic cups and macropinosomes.

Compiled Recommendations for the authors from the three Reviewers:

1. This paper would benefit from the inclusion of some evidence confirming the role of CYRI-B in human PDAC cells, as well as the inclusion of 3D invasion assays.

2. It would be good to discuss the distribution of CYRI-B in the tumours. The Lox-Cre system used is epithelial-specific, however, from the staining, it looks like CYRI-B is also expressed in the stroma. Is there anything known about the potential roles of CYRI-B in the stroma, as this plays a key role in PDAC progression?

3. Figure 1 shows levels of CYRI-B mRNA during PDAC progression. It would be good to also show stainings/quantifications of the protein itself.

4. The authors correctly use 70kDa dextran to visualize macropinocytosis. The results would be further strengthened by inhibiting micropinocytosis, possibly by depletion of the G protein-coupled receptor CaSR (Canton, Frontiers Immunol, 2018).

5. The authors use an inhibitor against LPAR1 and -3, but focus on LPAR1. This is based on their earlier results (Juin et al., 2019), where they showed that depletion of LPAR1, but not of LPAR3 reduces chemotaxis. This should be mentioned explicitly for a better understanding of the rationale.

6. The hypothesis is that depletion of CYRI-B would promote localized Rac1 activation at the membrane. However, the authors show that CYRI-B is found overexpressed in PDAC, consistent with other papers where its high expression correlates with poor outcome of many cancers. The prediction is that Rac1 functions modulated by CYRI-B would be inhibited in those tumours where CYRI-B is overexpressed. Is this the case and has it been formally demonstrated?

7. Most experiments use the depletion of CYRI-B to probe its function. It would be useful to readers and important to elaborate on how the specific CYRI-B functions shown upon depletion would fit with the in vivo observation of CYRI-B overexpressed in tumours. For example, loss of CYRI-B reduces chemotaxis potential. How can this result be reconciled with the predicted increase in Rac1 activation in the absence of CYRI-B? Conversely, a prediction of CYRI-B overexpression in human tumours would imply the inactivation of Rac1 whereas chemotaxis is promoted. The discussion could be improved with the addition of the authors' views and further explanations in this context.

8. Similarly, it is confusing to extrapolate a proposed increase in LPAR1 internalization by macropinocytosis with CYRI-B overexpression in PDAC. It is predicted that Rac1 would be locally inhibited in this scenario, and thus micropinocytosis would be compromised. It will be good to spell out what the authors envisage happens. For example, uptake could be switched to another receptor uptake process that would not involve CYRI-B sequestration of Rac1. Discussion of the potential alternatives will strengthen the manuscript.

9. "...LPAR1 is a cargo of CYRI-B dependent macropinocytosis" (page 21). This statement reads as an overinterpretation of the specificity of the process. It may suggest that there is a cargo selectivity by CYRI-B, which has not been formally demonstrated or is well accepted. Macropinocytosis is thought to occur as a bulk engulfment of the membrane and thus any receptor at the cell surface would be internalised non-specifically. The demonstrated reduction in LPAR1 uptake could be proportional to the interference with micropinocytosis rate by CYRI-B depletion for example.

10. Readers would benefit from more clear explanations of the differences and similarities between CYRI-A and CYRI-B. It will be important to clarify the specificity of the proposed functions of each protein. Both localize at the macropinosomes, modulate engulfment and regulate integrin a5b1 trafficking. It will be informative to specify if CYRI-A is also upregulated in human tumours, has a similar outcome as CYRI-B in vivo and also regulates LPAR1 uptake.

11. Upon depletion of CYRI-B in pancreatic tumour cells in vivo, the presence of similar levels of jaundice is confusing. Less metastasis is detected in the mesentery. Are liver metastasis affected in the absence of CYRI-B?

12. It would be helpful to improve further discussion of the data in the context of human tumours that contain CYRI-B overexpression and refine the interpretation of the results as described in the public review. The addition of alternative explanations for the cargo specificity of macropinocytosis and Rac1-independent alternatives as outlined above will go a long way to strengthen the manuscript.

13. It is important to demonstrate that an irrelevant surface receptor is not affected by CYRI-B depletion or the rate of reduced macropinocytosis. Revise or tone down the conclusion of specificity accordingly.

14. LPAR1 should be spelt out in the abstract.

---

## [Author Response]

Essential revisions (for the authors):The manuscript has been improved but there are some remaining issues that need to be addressed, as outlined below. Besides these comments, a number of suggestions have been made by the reviewers, which appear following the public reviews; the authors are encouraged to address as many of these as possible:1. The authors argue that CYRI-B could act as a buffer for Rac1 hyperactivation. This is likely based on earlier work of the authors using MDCK cells. However, no data on Rac1 activation and its regulation by CYRI-B in PDAC are shown. Please include respective data or discuss the possible, also alternative, molecular roles for CYRI-B in PDAC progression in more detail.

We agree with the comments and we have added some discussion (second paragraph of discussion) about this and clarified how we think this might be an advantage for tumours. We have also discussed the RAC1-FRET reporter mouse, which would be an interesting model to measure RAC1 activity in tumours with CYRI-B knockout. However, this would require a very large amount of mouse breeding and might take more than a year, so we feel that this experiment is beyond the scope of the current study.

2. It is not clear why the authors use COS cells for showing the role of CYRI-B in macropinocytosis, as also some data using CKPC cells are shown, which should be more relevant. For example, Figure 7B is the only figure showing CYRI-B-GFP and LPAR-1-mCherry on macropinosomes. However, this gallery does not show the beginning of the process. It would be important to show in CKPC cells that both proteins are already present at nascent macropinocytic cups that then lead to fully formed macropinosomes that are also positive for Rab5. So far, these results are a mosaic of data from COS and CKPC cells.

We have used COS7 cells for some of the experiments, because PDAC cells do not tolerate transfection of multiple probes very well and they are generally more difficult to work with. We have, however, tried to perform all of the essential experiments as much as possible in PDAC cells. We have repeated more of the experiments now in PDAC cells, as requested, so we hope that the reviewers will agree that the major points are supported better now with PDAC cell experiments. Please refer to the new Figure S6 and Video S5, where we have now shown CYRI-B localisation to macropinocytic structures and co-localisation with Rab5a in human pancreatic cancer AsPC-1 cells. Please see also the new Figure 7A and Video S6-S7, with additional data showing CYRI-B and Rab5a recruitment to macropinocytic structures in human pancreatic cancer AsPC-1 cells.

3. Related to the previous point: it would also be important to detect endogenous CYRI-B and LPAR1 in PDAC cells and to show their localization at macropinocytic cups and macropinosomes.

Unfortunately, we have not been able to detect either endogenous LPAR1 or CYRI-B in cells using immunofluorescence, despite multiple attempts with different antibodies. Because we have knockout cells, we can verify whether the antibodies work to a higher standard than previous studies, and we do not find any antibodies, including commercially available antibodies, that give convincing results for these experiments yet. This is definitely something that we agree we must do, but it awaits better reagents. This is also why we used in situ RNA-scope for our tumour study. We now mention this in the discussion.

Compiled Recommendations for the authors from the three Reviewers:1. This paper would benefit from the inclusion of some evidence confirming the role of CYRI-B in human PDAC cells, as well as the inclusion of 3D invasion assays.

We have now added new experiments, in Figure 7A, Figure S6 and Videos S5 and S6. We acknowledge that it is desirable to perform as many of the experiments as possible with human PDAC cells, but as these cells are a lot more difficult to transfect and work with, we have also supplemented these experiments with some mechanistic experiments in COS-7. We hope that the reviewer will agree that we now provide good evidence for the most crucial experiments in human PDAC cells.

2. It would be good to discuss the distribution of CYRI-B in the tumours. The Lox-Cre system used is epithelial-specific, however, from the staining, it looks like CYRI-B is also expressed in the stroma. Is there anything known about the potential roles of CYRI-B in the stroma, as this plays a key role in PDAC progression?

CYRI-B might be expressed in the stroma, but we do not have much evidence for or against this. In Figure 1B, we show CYRI-B expression in KPC tumours at the RNA level. Then in the CKPC, there is not much positive staining left after knockout in the epithelia- also panel B. So we think it is preliminary to discuss this, although we agree it is a very interesting point.

3. Figure 1 shows levels of CYRI-B mRNA during PDAC progression. It would be good to also show stainings/quantifications of the protein itself.

We agree, but we were unable to do this due to a lack of suitable antibody. We have added a comment on this in the discussion.

4. The authors correctly use 70kDa dextran to visualize macropinocytosis. The results would be further strengthened by inhibiting micropinocytosis, possibly by depletion of the G protein-coupled receptor CaSR (Canton, Frontiers Immunol, 2018).

We think that this experiment is beyond the scope of our study. We lack reagents to determine whether our cells express this GPCR or to deplete it.

5. The authors use an inhibitor against LPAR1 and -3, but focus on LPAR1. This is based on their earlier results (Juin et al., 2019), where they showed that depletion of LPAR1, but not of LPAR3 reduces chemotaxis. This should be mentioned explicitly for a better understanding of the rationale.

We have added some sentences on Page 14 to explain this better. We thank the reviewer for pointing this out.

6. The hypothesis is that depletion of CYRI-B would promote localized Rac1 activation at the membrane. However, the authors show that CYRI-B is found overexpressed in PDAC, consistent with other papers where its high expression correlates with poor outcome of many cancers. The prediction is that Rac1 functions modulated by CYRI-B would be inhibited in those tumours where CYRI-B is overexpressed. Is this the case and has it been formally demonstrated?

We agree that this is an interesting point to discuss. We think of it as a tug of war, where active KRAS drives RAC1 levels above what is healthy for the cells and CYRI-B may compensate and normalise RAC1. However, we do not have direct evidence for this, so we have toned this down and added some discussion around it in the Discussion section.

7. Most experiments use the depletion of CYRI-B to probe its function. It would be useful to readers and important to elaborate on how the specific CYRI-B functions shown upon depletion would fit with the in vivo observation of CYRI-B overexpressed in tumours. For example, loss of CYRI-B reduces chemotaxis potential. How can this result be reconciled with the predicted increase in Rac1 activation in the absence of CYRI-B? Conversely, a prediction of CYRI-B overexpression in human tumours would imply the inactivation of Rac1 whereas chemotaxis is promoted. The discussion could be improved with the addition of the authors' views and further explanations in this context.

We thank the reviewers for these suggestions. We have now added more discussion around these points in the Discussion section and we hope that the reviewers will find this clearer.

8. Similarly, it is confusing to extrapolate a proposed increase in LPAR1 internalization by macropinocytosis with CYRI-B overexpression in PDAC. It is predicted that Rac1 would be locally inhibited in this scenario, and thus micropinocytosis would be compromised. It will be good to spell out what the authors envisage happens. For example, uptake could be switched to another receptor uptake process that would not involve CYRI-B sequestration of Rac1. Discussion of the potential alternatives will strengthen the manuscript.

Thank you for the suggestion- we have also discussed these points in our revised discussion. It is true that the mechanisms behind some of these phenotypes are still unknown and require more study. We think that enhanced CYRI-B expression likely buffers the exacerbated Rac1 activation in KRAS-driven tumours, but this awaits further mechanistic studies.

9. "...LPAR1 is a cargo of CYRI-B dependent macropinocytosis" (page 21). This statement reads as an overinterpretation of the specificity of the process. It may suggest that there is a cargo selectivity by CYRI-B, which has not been formally demonstrated or is well accepted. Macropinocytosis is thought to occur as a bulk engulfment of the membrane and thus any receptor at the cell surface would be internalised non-specifically. The demonstrated reduction in LPAR1 uptake could be proportional to the interference with micropinocytosis rate by CYRI-B depletion for example.

We have softened this statement to acknowledge that other receptors likely also traffic by macropinocytosis. In fact, we have previously also shown that integrins can be cargoes and we have unpublished evidence that EGFR can be a cargo. We had not meant to imply that this was specific for LPAR1.

10. Readers would benefit from more clear explanations of the differences and similarities between CYRI-A and CYRI-B. It will be important to clarify the specificity of the proposed functions of each protein. Both localize at the macropinosomes, modulate engulfment and regulate integrin a5b1 trafficking. It will be informative to specify if CYRI-A is also upregulated in human tumours, has a similar outcome as CYRI-B in vivo and also regulates LPAR1 uptake.

We have added some discussion of CYRI-A. There is no evidence that CYRI-A is involved in human pancreatic tumours, but may be involved in renal and urothelial cancers, which we now clarify on page 23.

11. Upon depletion of CYRI-B in pancreatic tumour cells in vivo, the presence of similar levels of jaundice is confusing. Less metastasis is detected in the mesentery. Are liver metastasis affected in the absence of CYRI-B?

We agree that these are interesting questions. Probably the jaundice is caused when the bile duct gets blocked and this might be a stochastic process depending on where the metastatic nodules grow. We didn’t have enough mice that displayed liver metastasis in the transplant model to quantify this, but for the KPC model, we did show a reduction in liver metastasis. It would be interesting to check in a spleen to liver model, but we don’t feel that this use of more mice is currently justified.

12. It would be helpful to improve further discussion of the data in the context of human tumours that contain CYRI-B overexpression and refine the interpretation of the results as described in the public review. The addition of alternative explanations for the cargo specificity of macropinocytosis and Rac1-independent alternatives as outlined above will go a long way to strengthen the manuscript.

We have substantially revised the discussion and we hope that it is clearer now.

13. It is important to demonstrate that an irrelevant surface receptor is not affected by CYRI-B depletion or the rate of reduced macropinocytosis. Revise or tone down the conclusion of specificity accordingly.

We have added statements to tone this down. We did not mean to imply that the macropinocytosis was specific for LPAR1- see also comment above.

14. LPAR1 should be spelt out in the abstract.

Done.